# Genetic integration of behavioural and endocrine components of the stress response

**Thomas M Houslay[1]\*, Ryan L Earley[2], Stephen J White[1], Wiebke Lammers[1], Andrew J Grimmer[1], Laura M Travers[1†], Elizabeth L Johnson[2‡], Andrew J Young[1], Alastair Wilson[1]**

[1]Centre for Ecology and Conservation, University of Exeter (Penryn Campus), Penryn, United Kingdom; [2]Department of Biological Sciences, University of Alabama, Tuscaloosa, United States

**\*For correspondence:**
houslay@gmail.com

**Present address:** †School of Biological Sciences, University of East Anglia, Norwich, United Kingdom; ‡Southern Research, Birmingham, United States

**Competing interest:** The authors declare that no competing interests exist.

**Abstract** The vertebrate stress response comprises a suite of behavioural and physiological traits that must be functionally integrated to ensure organisms cope adaptively with acute stressors. Natural selection should favour functional integration, leading to a prediction of genetic integration of these traits. Despite the implications of such genetic integration for our understanding of human and animal health, as well as evolutionary responses to natural and anthropogenic stressors, formal quantitative genetic tests of this prediction are lacking. Here, we demonstrate that acute stress response components in Trinidadian guppies are both heritable and integrated on the major axis of genetic covariation. This integration could either facilitate or constrain evolutionary responses to selection, depending upon the alignment of selection with this axis. Such integration also suggests artificial selection on the genetically correlated behavioural responses to stress could offer a viable non-invasive route to the improvement of health and welfare in captive animal populations.

## Editor's evaluation

This is a timely paper on the genetic integration of behavioral and physiological components of the stress response in guppies. Using evolutionary quantitative genetic approaches, the authors show that genetic variation in the cortisol stress response is associated with genetic variation in stress-related behaviors. This result suggests that physiological and behavioral responses to stress should show correlated evolution in response to natural selection, which is of interest to evolutionary biologists and for animal welfare. The revised manuscript fully addresses both conceptual and methodological limitations of the earlier submission. Congratulations on a nice contribution to the literature.

## Introduction

Stress responses comprise suites of physiological and behavioural traits that enable individuals to cope with adverse conditions (*Romero, 2004*; *Overli et al., 2007*; *McEwen and Wingfield, 2010*; *Taborsky et al., 2021*). Some individuals are likely better at coping with adverse conditions than others, and understanding the role played by underlying genetic variation could have important implications for managing stress-related disease in captive populations and predicting the evolutionary responses of free-living populations to both natural and anthropogenic stressors (*Barton and Iwama, 1991*; *Koolhaas et al., 1999*; *McEwen and Wingfield, 2003*; *Romero, 2004*; *Koolhaas, 2008*). For instance, if among-individual differences in stress response traits are a product of genetic variation (*Koolhaas et al., 1999*; *Koolhaas et al., 2007*) then they may be a viable target for artificial

selection strategies (*Mignon-Grasteau et al., 2005*). This could be used to reduce stress-related welfare issues in captive populations (e.g. in livestock; *Broom and Johnson, 1993*; *von Borell, 1995*; *Möstl and Palme, 2002*; *Kasper et al., 2020*). In free-living populations, variation in stress response traits is expected to cause fitness variation under stressful conditions (*Wingfield, 2003*; *Koolhaas, 2008*). Thus, exposure to stressors could lead to evolutionary changes in the distributions of traits that contribute to a population's long-term resilience in the face of natural (e.g. predation risk; *Clinchy et al., 2013*) and/or anthropogenic (*Tarlow and Blumstein, 2007*; *Busch and Hayward, 2009*; *Angelier and Wingfield, 2013*; *Sadoul et al., 2021*) challenges. However, as the evolutionary response of any trait to selection is determined in large part by its genetic variation, our ability to predict – and potentially harness – evolutionary changes in the stress response is currently hampered by limited understanding of the underlying genetics.

Natural selection does not act on single traits in isolation, but rather on multivariate phenotypes (*Lande and Arnold, 1983*; *Blows, 2007*). This is likely to be an important consideration for understanding the evolution of the stress response. For instance, while glucocorticoid (GC) levels are frequently used to measure the stress response (*McEwen and Wingfield, 2003*; *Korte et al., 2005*), an individual's first line of defence against acute environmental challenges will typically be behavioural (*Moberg, 2000*). This may include risk avoidance strategies as well as the widely known 'fight-or-flight' responses. Subsequent GC release then serves to mediate physiological (and further behavioural) responses (*Wingfield et al., 1998*; *Wingfield and Kitaysky, 2002*). Natural selection is therefore expected to favour combinations of behavioural and physiological stress response traits that act synergistically to maintain fitness under stressful conditions (*Koolhaas et al., 1999*; *Overli et al., 2007*). Evolutionary theory predicts that correlational selection will shape the structure of multivariate quantitative genetic variance (as represented by the genetic covariance matrix **G**; *Cheverud, 1982*; *Blows, 2007*; *Roff and Fairbairn, 2012*), contingent on the distribution of new pleiotropic mutations that generate multivariate genetic variance as well as the selection that depletes it (*Blows and Walsh, 2009*; *Walsh and Blows, 2009*). In general, correlational selection should (under certain assumptions) have direct and strong effects on genetic covariances (*Lande, 1980*; *Jones et al., 2003*; *Arnold et al., 2008*). In the context of the stress response, we should therefore expect genetic – as well as phenotypic – integration of behavioural and physiological traits (*McGlothlin and Ketterson, 2008*; *Ketterson et al., 2009*; *Cox et al., 2016*). By genetic integration, we mean genetic correlation structure among suites of traits. Genetic integration is hypothesised to underpin adaptive variation in multivariate phenotypes in many fields of evolutionary biology (e.g. life history [*Stearns, 1989*; *Roff, 1992*], behavioural ecology [*Sih and Bell, 2008*]), but has not been tested for explicitly among stress response components using quantitative genetic approaches.

The most compelling evidence for genetic integration of behavioural and physiological stress response traits to date comes from artificial selection experiments on domestic animal populations (e.g. rainbow trout *Oncorhynchus mykiss* (*Pottinger and Carrick, 1999*), Japanese quail *Coturnix coturnix japonicus* (*Jones et al., 1994*), house mice *Mus musculus domesticus Veenema et al., 2003*). For example, lines of rainbow trout selected for stress-induced plasma cortisol levels (*Pottinger and Carrick, 1999*) experienced correlated evolutionary changes in behaviour (*Overli et al., 2005*). In a rare study of a wild-type population (albeit under captive conditions), cortisol levels were found to evolve in response to selection on behavioural 'personality' in great tits (*Parus major*; *Carere et al., 2003*). In these examples, the correlated responses of behavioural and physiological stress response traits to selection are consistent with some degree of genetic integration of these behavioural and physiological traits. However, the extent and 'structure' of this integration remains unclear, and some results were inconsistent with a hypothesised simple axis of genetic (co)variation among behavioural and physiological components of the stress response. For example, in the trout study, the 'low-cortisol response' selected lines actually showed a higher metabolic stress response under confinement (suggestive of opposing responses to selection by different physiological components of the stress response; *Trenzado et al., 2003*). In the same selection lines, there was also no link between 'boldness' (an aspect of personality related to risk-taking behaviour) and cortisol response under standardised testing (*Thomson et al., 2011*).

While selection experiments illustrate that genetic integration of behaviour and physiology can occur, estimation of the genetic variance-covariance matrix (**G**) through quantitative genetic modelling provides a complementary strategy that also allows investigation of exactly how multivariate

genetic variation is structured within populations. In the context of the stress response, this should provide insights into both how selection has acted in the past (*Ketterson et al., 2009*), and whether responses to future selection are likely to be constrained (*Blows and Walsh, 2009*; *Walsh and Blows, 2009*). Insight into past selection follows from the fact that strong correlational selection should lead to integration of traits in **G** over the long term, a phenomenon explored most commonly for suites of morphological traits (following, e.g. *Cheverud, 1982*), but that is equally applicable to any aspect of the phenotype (see, e.g. *Hine et al., 2004*; *Hunt et al., 2007*; *Oswald et al., 2013* for examples pertaining to behavioural evolution and mate choice). Insight into future responses to selection follows from the fact that the direction (in multivariate trait space) and magnitude of a response to selection is limited by the amount of variance in **G** that is in alignment with the vector of (directional) selection β (*Blows and Walsh, 2009*; *Walsh and Blows, 2009*). Integration in **G** is manifest as genetic correlations between trait pairs, and also at the level of the whole matrix by an overdispersion of eigenvalues—indicating that there are fewer effective dimensions of genetic variation than there are traits measured (*Blows, 2007*).

Here, we estimate **G** for behavioural and physiological components of the acute stress response in Trinidadian guppies (*Poecilia reticulata*). This enables us to determine not only (i) whether these components are genetically integrated on the major axis of genetic (co)variation (i.e. the first eigen vector of **G**, $\mathbf{g}_{max}$), but also (ii) whether the structure and orientation of this axis suggests variation in overall stress responsiveness and/or 'coping style' (explained further below; *Koolhaas et al., 2010*; *Boulton et al., 2015*). We use fish from a captive colony of guppies derived from wild ancestors sampled from the Aripo River, Trinidad in 2008 and subsequently maintained at high population size (with no deliberate inbreeding or selection). We have validated the use of standardised 'open field trials' (OFTs) for testing (acute) behavioural stress responses in this species (*Houslay et al., 2018*), and demonstrated significant additive genetic (co)variance underpinning variation in risk-taking, exploratory, and 'flight' type components of the behavioural stress response using this testing paradigm (*White et al., 2018*; *White and Wilson, 2019*). We have also demonstrated, using a non-invasive waterborne hormone sampling method, that individuals differ significantly in their GC (specifically, free circulating cortisol) response to an acute stressor (handling, coupled with short-term isolation and confinement; *Houslay et al., 2019*) and that, on average, this physiological response declines with repeated stressor exposure (consistent with habituation). Nothing is known about the genetic basis of variation in these physiological traits, or about their integration (phenotypically or genetically) with behavioural components of the stress response.

First, we combine OFT results with complementary 'emergence trials' (ET) and 'shoaling trials' (ST) to characterise among-individual and genetic variation in the behavioural stress response. Second, we characterise the physiological stress response and its rate of habituation by assaying GC levels following first and third exposure to a handling and confinement stressor (see Materials and methods). Utilising repeated behavioural and physiological testing of individual fish within a known pedigree structure, we are able to estimate the repeatable (among-individual) component of phenotypic (co)variance in these stress response traits, and then determine the additive genetic contribution to this (**G**; the genetic variance-covariance matrix for this suite of stress response traits). We predicted that individual traits will be heritable and that **G** will contain strong genetic correlation structure between behavioural and physiological components of the stress response consistent with genetic integration. We also predicted that both behavioural and endocrine components of the stress response would load on the major axis of genetic variation in multivariate trait space, $\mathbf{g}_{max}$. The 'stress coping style' model (*Koolhaas et al., 1999*) predicts variation in the type of response to stress. Simplistically, as originally proposed this verbal model posits that individuals (or genotypes) perceive equal degrees of stress but differ in how this manifests phenotypically: genotypes at one end of the axis having 'reactive' behavioural phenotype (e.g. freezing behaviour) coupled to lower GC levels, while the 'proactive' end is characterised by more active 'fight or flight' behaviour coupled to higher GC levels. However, previous analyses of this population suggest variation may be more in terms of 'stress responsiveness' than stress coping style (*Houslay et al., 2018*; *Prentice et al., 2020*; *White et al., 2020*). That is to say, some individuals (or genotypes) perceive the trial as a more severe stimulus and exhibit more characteristic stress behaviours (e.g. flight and/ or freezing, thigmotaxis) while others show more typical 'unstressed' behavioural profiles (e.g. exploration of the arena). In this scenario, we predicted high GC levels to co-occur with characteristic stress behaviours, and low GC levels with 'unstressed' behavioural profiles.

## Results

In total, we obtained multivariate behavioural data from 5966 trials (3379 OFTs, 1548 ETs and 1039 STs) on 1384 individual fish. The number of individuals phenotyped (OFTs = 1365, ETs = 806, STs = 532) and the mean number of observations per fish (OFTs = 2.5, ETs = 1.9, STs = 2.0) varied across the behavioural data types. All fish were contained within a genetic pedigree structure comprising maternal full-sibships nested within paternal half-sibships. This structure was produced via multiple rounds of breeding work and has a maximum depth of five 'generations'. Some of the OFT data have already been used in studies of the evolutionary genetics of personality (*White et al., 2018*; *White and Wilson, 2019*), but here we extend that dataset and use it in conjunction with other behavioural and physiological measures for different purposes. We also obtained 1,238 waterborne assays of cortisol levels for 629 fish (almost all from the final generation). The handling and confinement stressor applied for this assay was performed three times (at 48 hr intervals) for all fish tested, but the holding water sample was only processed for GC content at two time points (the first and last confinement, subsequently $Cortisol_1$ and $Cortisol_3$). Full details of husbandry, phenotyping and analysis are provided in Materials and methods.

### Genetic variance in behavioural components of the stress response

Behavioural data were extracted from OFTs, ETs, and STs using video tracking of fish (as described in *White et al., 2016*; *Houslay et al., 2018*). Time to emerge from the shelter ('*emergence time*') was extracted from ETs and natural log (ln) transformed for analysis, while *shoaling tendency* was calculated from STs as the time spent in the third of the tank closest to a same-sex shoal (which was visible but physically separated) minus the time spent in the third of the tank farthest from the shoal. The OFT, ET, and ST testing paradigms are all considered to assay behavioural components of the stress response in the broad sense, as each test starts with the capture and transfer of the focal fish into a novel and brightly lit arena away from their home tank and familiar tank mates.

Four traits were defined from the OFT and measured by videotracking for 4 m 30 s after an initial 30 second acclimation period upon transfer into the arena: *track length* (distance swum), *area covered* (as a proportion of the arena floor area), the number of *freezings* (the number of times a fish's speed dropped below the minimum velocity threshold of 4 cm/s for at least 2.5 s, *Houslay et al., 2018*; this trait was square root transformed for analysis) and time in the middle (i.e. in the central area of the open field arena away from the tank walls, which is assumed to be perceived as riskier, e.g. *Houslay et al., 2018*). Note low values of *time in the middle* imply thigmotaxis (i.e. tendency to avoid exposure to potential threats by hugging walls), and were very highly correlated with a measure of average distance from the wall at the observation level ($r = 0.94$, $t_{1,3368} = 159.2$, $p < 0.0001$; see Appendix 1 for further discussion of the selection of OFT behavioural traits). All these OFT traits are repeatable and heritable in this population (*White et al., 2016*; *White et al., 2018*; *Houslay et al., 2018White and Wilson, 2019*).

The absence of a strong positive correlation between *track length* and *area covered* (*Figure 1A*) is notable and potentially biologically informative; if fish moved randomly with respect to direction in the arena then *area covered* would increase monotonically (to an asymptote at 100%) with *track length*. A possible explanation is that a long *track length* arises sometimes from a (putatively) less stressed fish exploring the arena (fish one in *Figure 1B*) and sometimes from a (putatively) more stressed fish exhibiting a typical 'flight' response (fish four in *Figure 1B*). These two types of response can be discriminated based on whether, in a given trial, higher track length is associated with higher *area covered* and *time in the middle* (exploration) or the converse (flight response).

To quantitatively discriminate between exploratory behaviour and flight responses we derived a new trait, '*relative area covered*'. We used a simple simulation procedure (see Materials and methods) to predict *expected area covered* for any given *track length* under a null 'random swim' within the arena (*Figure 1C*). *Relative area covered* is then calculated as observed *area covered* – *expected area covered* given the *track length* (*Figure 1D*) and will be high for fish engaging in exploration, and low for an obvious 'flight' response manifest as rapid swimming around the tank walls. Our subsequent analyses then used this derived *relative area covered* trait in place of *area covered*.

Pedigree-based 'animal models' (*Wilson et al., 2010*) were used to test for and estimate additive genetic variation in each of the six behavioural traits (*emergence time*, *shoaling tendency*, and the four OFT traits) while controlling statistically for social housing group and non-genetic sources of

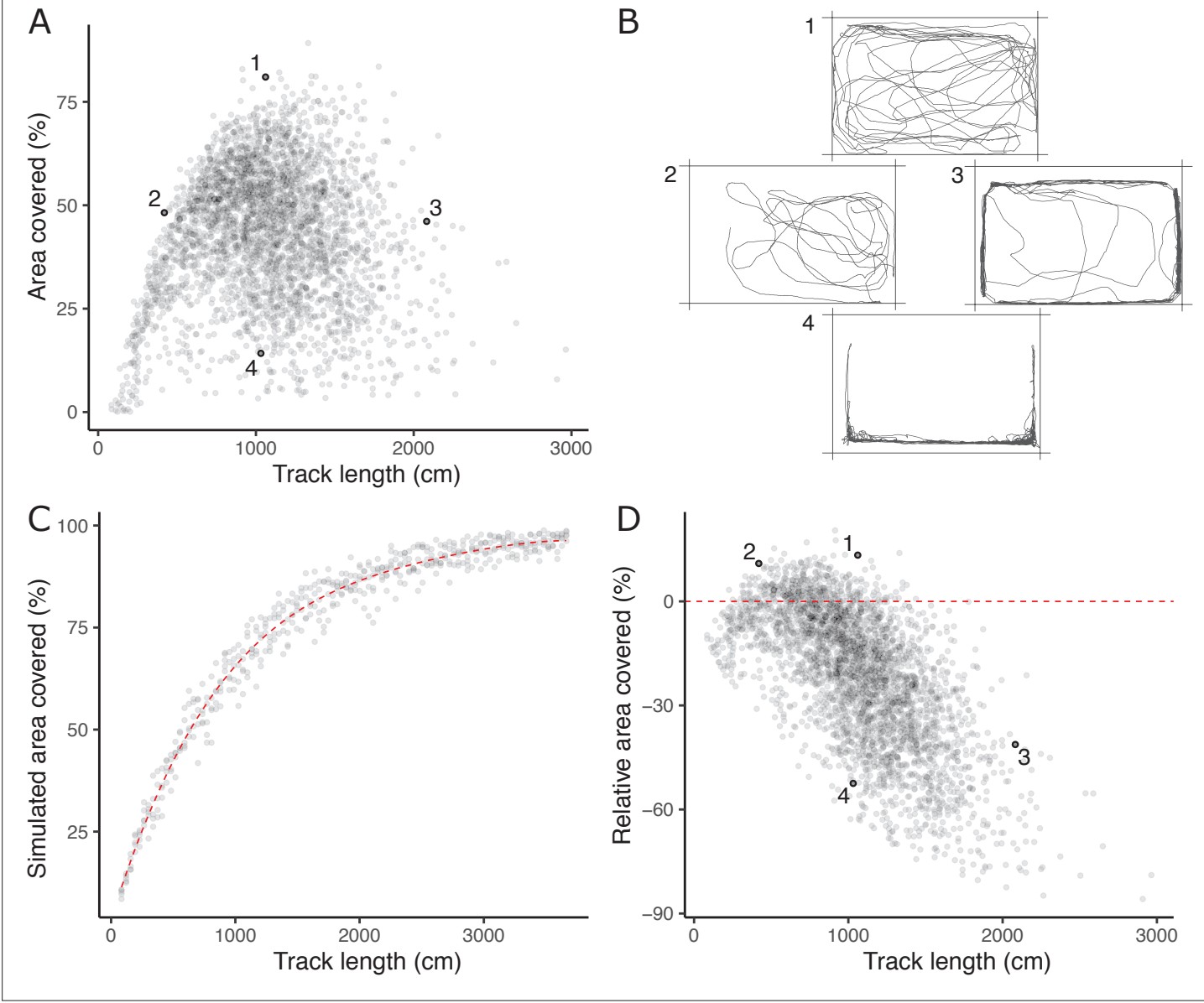

**Figure 1.** The lack of a strong positive relationship between observed *track length* and *area covered* (panel **A**), is initially puzzling given expected autocorrelation and that both are used as positive indicators of exploratory (or 'bold') behaviour. Inset examples of OFT tracks from four individuals (panel **B**) shed light on this. Fish 1 and 2 appear to be exploring the tank, while 3 and 4 are engaging in stereotypical 'flight' behaviour characterised by strong thigmotaxis (remaining close to tank walls) and/or rapid movement along tank walls. As a consequence, individuals 2 and 3 have similar *area covered* during the OFT, but very different *track lengths*. We simulated random movements to define an expected null relationship between *area covered* and *track length* (panel **C**; dashed red line shows the fourth order polynomial model fit; see Materials and methods). The polynomial regression was then used to predict the expected area covered under random movement for each trial's observed *track length*, and the '*relative area covered*' was calculated as the observation minus this prediction. Panel **D** shows the resultant *relative area covered* plotted against *track length* for all trials (dashed red line at *relative area covered* = 0, shows where individuals of any *track length* are expected to lie if they move randomly with respect to direction). From this it is apparent that fish 1 and 2 have high *relative area covered*, while 3 and 4 do not.

among-individual variance (as well as several fixed effects; see Materials and methods for full details). These confirmed the presence of significant additive genetic variation for the *relative area covered* trait, as well as for *track length*, *time in the middle* (as expected from previous findings; *White et al., 2018*; *White and Wilson, 2019*), √*freezings* and *ln emergence time* (*Table 1*). With the exception of *shoaling tendency*, heritabilities (conditional on fixed effects; see Materials and methods) are low to moderate (range of 9–20%; *Table 1*) and within the expected range for behaviours (*Stirling et al.,*

**Table 1.** Estimated variance components, along with adjusted heritability, for each trait as estimated in a univariate model (± standard error).
Chi-square test statistics and p-values are provided for the pedigree term, testing for the presence of significant additive genetic variance ($V_a$).

| Trait | $V_a$ | $V_{pe}$ | $V_{group}$ | $V_{residual}$ | $h^2$ | $\chi^2_{0,1}$ | p |
|---|---|---|---|---|---|---|---|
| *Relative area covered* | 42.87 ± 12.89 | 56.16 ± 10.92 | 19.17 ± 4.92 | 203.17 ± 6.26 | 0.13 ± 0.04 | 31.2 | < 0.001 |
| *Time in the middle* | 554.78 ± 144.18 | 473.24 ± 114.77 | 196.94 ± 53.54 | 2002.06 ± 62.38 | 0.17 ± 0.04 | 47.2 | < 0.001 |
| *Track length* | 23584.54 ± 5534.90 | 26587.74 ± 4556.87 | 3173.34 ± 1396.84 | 76140.49 ± 2376.32 | 0.18 ± 0.04 | 88.7 | < 0.001 |
| *√Freezings* | 0.34 ± 0.08 | 0.18 ± 0.06 | 0.08 ± 0.02 | 1.13 ± 0.04 | 0.20 ± 0.05 | 48.2 | < 0.001 |
| *ln Emergence time* | 0.12 ± 0.05 | 0.06 ± 0.06 | 0.05 ± 0.02 | 1.07 ± 0.05 | 0.09 ± 0.02 | 22.5 | < 0.001 |
| *Shoaling tendency* | 0 ± 0 | 2457.36 ± 570.96 | 708.87 ± 316.30 | 9900.95 ± 622.10 | 0 ± 0 | 0 | 0.5 |
| *ln Cortisol* | 0.08 ± 0.02 | 0.02 ± 0.02 | 0.01 ± 0.01 | 0.15 ± 0.01 | 0.31 ± 0.09 | 24.4 | < 0.001 |

*2002*). We detected no additive genetic variance for *shoaling tendency* (*Table 1*), despite there being repeatable differences among individuals ($R = 0.19 ± 0.04$; $\chi^2_{0,1} = 20.01$, $p < 0.001$).

## Genetic variance in physiological components of the stress response

Using a series of nested bivariate animal models, we tested for additive genetic variance in cortisol levels (ln-transformed) following stressor exposure (handling and confinement) and for genotype-by-environment interaction (GxE). In this context, the environment (E) is the trial number in each fish's stress trial series (i.e., cortisol level after the first or third trial). Any GxE present can therefore be interpreted as genetic variance for habituation to the stressor, given that the average cortisol level was lower following exposure to the third stressor than the first (ln transformed ng/hr, mean ± SE; $Cortisol_1 = 8.47 ± 0.05$, $Cortisol_3 = 8.02 ± 0.05$, Wald $F_{1,38.8} = 110.0$, $p < 0.001$; see Materials and methods for explanation of units). We first modelled $Cortisol_1$ and $Cortisol_3$ as distinct response variables in a bivariate framework assuming no GxE (such that we constrain $V_{A-Cortisol1} = V_{A-Cortisol3}$ and the cross context additive genetic correlation $r_{A-Cortisol1,Cortisol3} = 1$). This model revealed a significant additive genetic component to variation among individuals in their cortisol levels following stressor exposure ($\chi^2_{0,1} = 18.2$, $p < 0.001$).

Expanding the model to allow GxE by estimating separate genetic variances for $Cortisol_1$ and $Cortisol_3$ provides a significantly better fit to the data ($\chi^2_{0,1} = 3.8$, $p = 0.03$), meaning GxE is present. This can be viewed as genetic variance for habituation to repeated stressor exposure, or as a change in genetic variance for cortisol from the first to the third sampling (*Figure 2*). These are two perspectives of the same phenomenon; a reduction in additive genetic variance between the first stressor ($V_{A-Cortisol1} = 0.081 ± 0.029$) and the third ($V_{A-Cortisol3} = 0.041 ± 0.025$) arises because genotypes with higher-than-average levels for $Cortisol_1$ habituate more rapidly (i.e. have more negative reaction norm slopes). Note however that allowing the cross-context genetic correlation to deviate from +1 does not significantly improve the model fit ($\chi^2_1 = 0.0$, $p = 1$): thus the rank order of the genotypes does not appreciably change across the two contexts (i.e. genetic reaction norms show little crossing; *Figure 2*), so there is a strong positive cross-context genetic correlation ($r_{A-Cortisol1,Cortisol3} ± SE = 0.83 ± 0.21$).

In this model we also find that variance in cortisol explained by housing group effects is similar across contexts ($V_{Group-Cortisol1} = 0.038 ± 0.014$, $V_{Group-Cortisol3} = 0.032 ± 0.012$), but that residual (unexplained) variance is greater after the third stressor exposure ($V_{R-Cortisol1} = 0.164 ± 0.022$, $V_{R-Cortisol3} = 0.237 ± 0.023$). In combination, the changes in both additive genetic and residual variance between the two contexts lead to appreciably higher heritability for cortisol levels following the first stressor exposure relative to the third ($h^2_{Cortisol1} = 0.285 ± 0.094$, $h^2_{Cortisol3} = 0.131 ± 0.078$).

## Testing for genetic integration and identifying the major axis of genetic (co)variance

There is strong evidence for phenotypic integration of *Cortisol* with behaviour at the among-individual levels (see *Supplementary file 5*). To test for and characterise the hypothesised genetic integration between behavioural and physiological components of the stress response, we built a multivariate

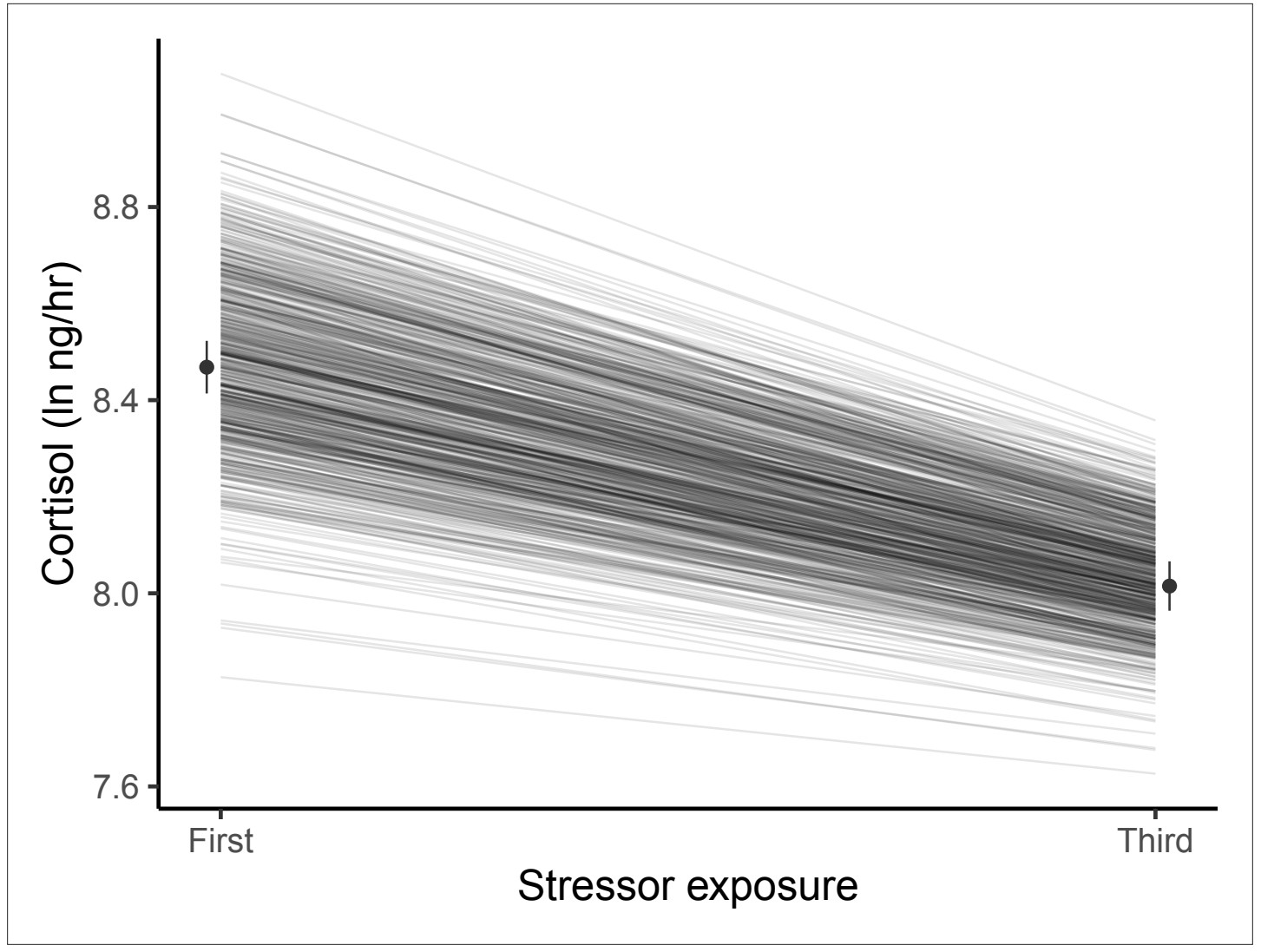

**Figure 2.** Guppies habituate to the waterborne sampling procedure, as shown by a decline in average ln-transformed cortisol level (ng/hr) following stressor exposure between first and third exposures. Black circles and associated bars denote predicted population means (± standard error) from mixed model analysis. Gray lines depict the predicted genetic reaction norms across repeated stressor exposure for each individual. Weak, but statistically significant GxE is manifest as variance in the genetic reaction norm slopes (i.e. lack of parallelism) and results in a slight reduction of genetic variance for cortisol at the third exposure relative to the first.

animal model to estimate **G**. We excluded *shoaling tendency* given the absence of detectable genetic variance in the univariate model. We also elected to treat cortisol as a single trait (allowing for a fixed effect of stressor exposure number [1 vs 3] on the mean). Although the above analysis demonstrates GxE for cortisol, the strong positive cross-context genetic correlation justifies collapsing $Cortisol_1$ and $Cortisol_3$ into a single trait to maximise statistical power to detect any genetic covariance with behaviour.

Our final model contained six response traits: *relative area covered, time in the middle, track length,* (square root-transformed) *freezings,* (ln-transformed) *emergence time,* and (ln-transformed) *Cortisol* (now treated as two repeats of a single trait). We standardised all traits to standard deviation units (after transformation where applicable) to assist multivariate model fitting and to prevent eigen-vectors of **G** (see below) being dominated by traits with higher variance in observed units. To simplify interpretation of **G,** we also multiplied *emergence time* by –1 after transformation. Thus, high values denote rapid emergence from the shelter.

The resultant estimate of **G** (*Table 2*) contains significant additive genetic covariance structure overall (Likelihood Ratio Test of the full model vs. a reduced model **G** that contains variances but

**Table 2.** Additive genetic covariance-correlation matrix (G) from the full multivariate animal model.
Genetic variances provided on the shaded diagonal, with genetic covariances below and genetic correlations above. 95% confidence intervals in parentheses are estimated from 5000 bootstrapped replicates. Where the confidence intervals for any estimate do not cross zero the estimate is considered statistically significant (at the 0.05 alpha level) and are shown in bold.

| | Relative area covered | Time in the middle | Track length | √Freezings | -ln Emergence time | Ln Cortisol |
|---|---|---|---|---|---|---|
| Relative area covered | 0.115 (0.050,0.182) | 0.795 (0.601,0.952) | −0.549 (-0.789,−0.239) | 0.139 (-0.260,0.511) | −0.438 (-1.305,0.239) | −0.376 (-0.959,0.272) |
| Time in the middle | 0.103 (0.044,0.165) | 0.145 (0.070,0.215) | −0.658 (-0.86,−0.414) | 0.363 (0.024,0.659) | −0.153 (-0.815,0.538) | −0.617 (-1.139,−0.155) |
| Track length | −0.071 (-0.127,−0.017) | −0.096 (-0.154,−0.035) | 0.147 (0.080,0.219) | −0.801 (-0.931,−0.647) | 0.61 (0.064,1.328) | 0.425 (-0.027,0.968) |
| √Freezings | 0.020 (-0.034,0.076) | 0.059 (-0.001,0.120) | −0.132 (-0.200,−0.060) | 0.185 (0.091,0.269) | −0.483 (-1.103,0.146) | −0.556 (-1.069,−0.067) |
| -ln Emergence time | −0.041 (-0.103,0.012) | −0.016 (-0.073,0.041) | 0.065 (0.008,0.121) | −0.057 (-0.126,0.009) | 0.076 (0.009,0.148) | −0.020 (-0.807,0.790) |
| ln Cortisol | −0.044 (-0.105,0.023) | −0.082 (-0.145,−0.018) | 0.057 (-0.002,0.119) | −0.083 (-0.155,−0.01) | −0.002 (-0.058,0.053) | 0.121 (0.036,0.206) |

not covariances: $\chi^2_{15}$ = 91.06, p < 0.001). We found a large number of strong genetic covariance/correlation estimates between trait pairs that were deemed statistically significant (based on the bootstrapped 95% confidence intervals not crossing zero). The 4 OFT traits (*relative area covered, time in the middle, track length,* and *√freezings*) show strong patterns of pairwise genetic correlation. *Track length* shows significant negative genetic correlation with *relative area covered, time in the middle,* and *√freezings* (such that genotypes predisposed to higher values of *track length* are also predisposed to lower values of *relative area covered, time in the middle,* and *√freezings,* and vice versa). *Relative area covered* and *time in the middle* show significant positive genetic correlation with one another, and *√freezings* shows significant positive correlation with *time in the middle*. The only genetic correlation among OFT traits that is not significant is between *√freezings* and *relative area covered*. The speed at which individuals emerge from the shelter during the emergence trials, *-ln emergence time*, shows significant positive genetic correlation with *track length* (i.e., genotypes predisposed to higher values of *track length* in the OFT are also predisposed to faster *emergence* in the ET). There are no significant pairwise genetic correlations between *–ln emergence time* and the other OFT traits, but all correlation estimates are negative. Pairwise genetic correlations between *ln cortisol* and all behavioural traits are illustrated in *Figure 3*. Ln-transformed *cortisol* shows significant negative genetic correlations with both *time in the middle* and *√freezings*. Genetic correlations between *ln cortisol* and other OFT traits were not significantly different from zero (based on 95% confidence intervals), with correlations estimated as negative with *relative area covered* and positive with *track length*. The genetic correlation between *–ln emergence time* and *ln cortisol* is also not significant, and is estimated at very close to zero.

Eigen decomposition of **G** provides a more holistic view of the genetic covariance structure and the level of integration among traits. Here, the major axis **g_max** (first principal component, PC1, with 95% confidence intervals from 5000 bootstrap replicates) explains 59.5% (47.5%, 75.7%) of the genetic variance in multivariate phenotype. Subsequent axes necessarily capture declining proportions of the multivariate genetic variance (PC2 = 20.2% [14.8, 32.1]; PC3 = 14.1% [5.6, 19.6]; PC4 = 4.5% [0.9, 8.2]; PC5 = 1.0% [0, 2.8], PC6 = 0.7% [0,0.1]; see *Supplementary file 6*, *Figure 4—figure supplement 1*). All traits except *-ln emergence time* load significantly on **g_max** (*Figure 4*). *Relative area covered, time in the middle* and *√freezings* load in one direction, while *track length* and *ln Cortisol* load in the other direction. This structure indicates a major axis of genetic variation in integrated stress response (*Figure 4—figure supplement 2*), where genotypes at one end of this axis can be considered to have a 'freeze' (or possibly 'weaker'; discussed later) type of behavioural stress response to the OFT assay (i.e. freezing more frequently, swimming shorter distances, spending more time in the central area of the tank, and exhibiting exploratory swimming patterns that cover greater areas relative to their distance swum). This behavioural profile is associated with 'weaker' physiological responses to stress

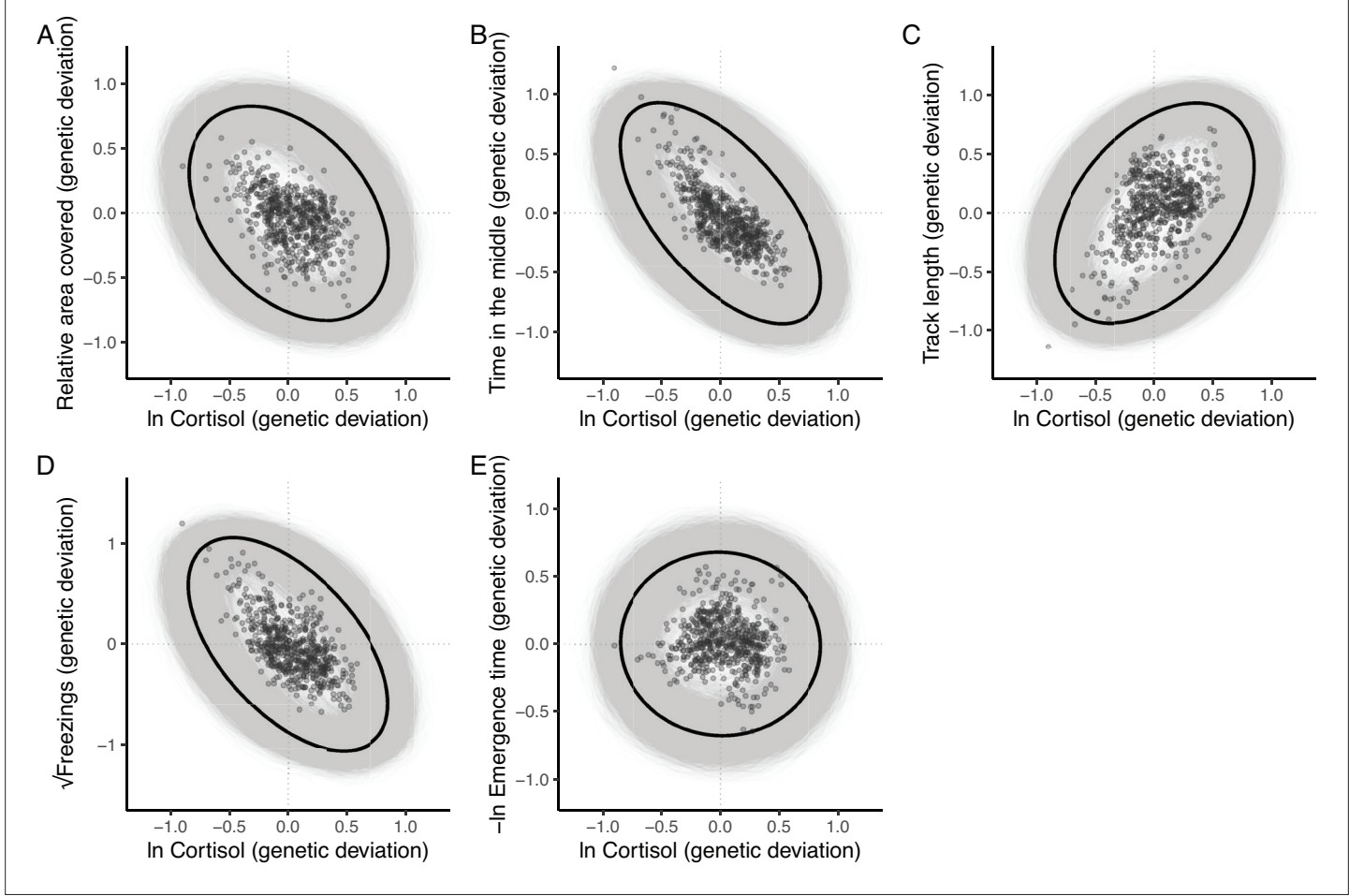

**Figure 3.** The additive genetic relationship between ln-transformed cortisol (x-axis) and five behaviours (a, *relative area covered*; b, *time in the middle*; c, *track length*; d, *√freezings*; e, *-ln emergence time*). Points show (predicted) bivariate genetic deviations from the population means, plotted for those individuals in the pedigree with cortisol data. In each case the black ellipse depicts the 'shape' of the relationship as given by the point estimate of G. Specifically it encompasses the area expected to contain 95% of the bivariate genetic distribution for the population. Grey ellipses denote the corresponding areas defined from 5000 bootstrapped replicates of G, and so highlight the uncertainty around these bivariate distributions.

(i.e. producing lower cortisol levels in response to the stressor). Meanwhile, genotypes at the other end of $\mathbf{g_{max}}$ have a more 'flight' (or arguably 'stronger') type of behavioural stress response to the OFT assay (i.e. freezing less frequently, swimming further, spending more time close to the tank edges, and covering less area relative to their distance travelled in OFTs). This behavioural profile is associated with 'stronger' physiological responses to stress (i.e. producing higher cortisol levels in response to the stressor).

## Discussion

In this study, we sought to determine whether – and to what extent – there exists genetic variation for, and integration between, behavioural and physiological (endocrine) components of the stress response. Our results provide three main novel insights. First, we find that genetic variation does underpin individual differences in both behavioural and physiological components of the stress response. Second, we find genetic covariance structure among these behavioural and physiological traits, indicating that they are indeed genetically integrated. Thirdly, having identified the structure of the major axis $\mathbf{g_{max}}$ of the genetic variance-covariance matrix **G** we suggest that it is more readily interpreted as an axis of genetic variation in stress coping style than in stress responsiveness, although we acknowledge that any distinction may be somewhat subjective. Overall, by estimating the genetic covariance structure

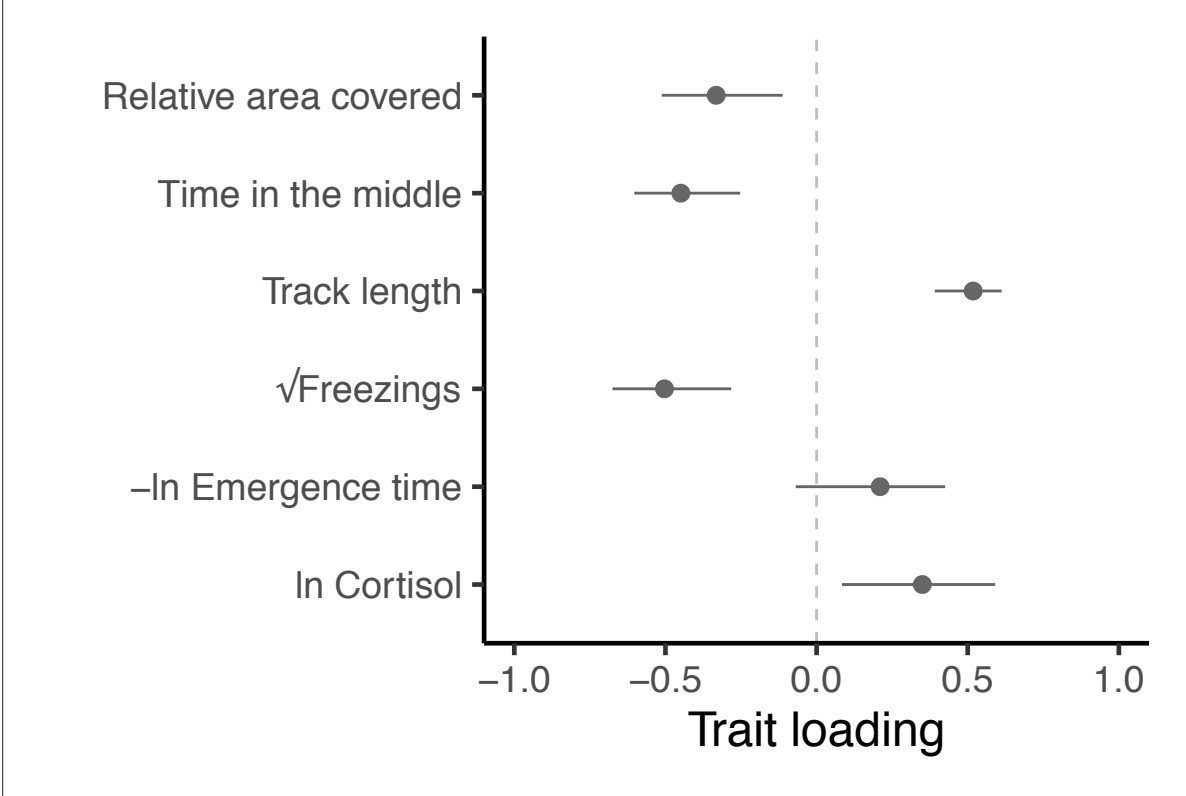

**Figure 4.** Trait loadings from $g_{max}$, the first eigen vector (principal component) of G. This axis explains 59.5% of the genetic (co)variation found in the focal behavioural and physiological components of the stress response in our guppy population. Points show trait loadings from the first eigen vector of our estimate of G, with bars representing 95% confidence intervals on each loading (calculated from 5000 bootstrapped replicates of the model).

The online version of this article includes the following figure supplement(s) for figure 4:

**Figure supplement 1.** Scree plot showing the proportion of total variance explained by each eigen vector, with vertical bars indicating 95% confidence intervals as estimated from 5000 bootstrapped replicates.

**Figure supplement 2.** Histogram of PC scores for $g_{max}$ calculated for each individual that was assayed for both behaviour and endocrine responses.

among traits we find the first quantitative genetic support to date for the hypothesis of evolutionary integration between behavioural and endocrine components of the stress response.

We find heritable variation in, and covariation among, behaviours assayed in the open field trial (OFT), including the derived trait *relative area covered*. The latter trait, derived by considering an appropriate biological null model of the relationship between *track length* and (absolute) *area covered*, serves as a useful proxy for exploratory behaviour. Here, we demonstrate an axis of repeatable and heritable variation that spans from 'freeze' type responses characterised by less active but more exploratory swimming patterns (more *freezings* and lower *track length*, coupled with higher *relative area covered* and *time in the middle*) through to a 'flight' type response characterised by higher activity coupled with thigmotaxis and low exploration (fewer *freezings* and higher *track length*, coupled with lower *relative area covered* and *time in the middle*). Given the wide use of OFTs in biomedical research (e.g., *Rex et al., 1998*; *Karl et al., 2003*) as well as in animal behaviour, our phenotyping approach (including the derivation of *relative area*) may have broad applicability for discriminating between exploration and stress/anxiety-related patterns of behaviour. The OFT paradigm is also widely applied to studies of 'shy—bold' type personality variation in fishes (*Toms et al., 2010*) and other vertebrates (*Carter et al., 2013*; *Perals et al., 2017*). The extent to which behavioural differences deemed characteristic of a 'shy—bold' personality axis (commonly, if not universally, defined as repeatable variation in response to perceived risk; *Sloan Wilson et al., 1994*) should be viewed as equivalent to variance in 'stress coping style' and/or 'stress responsiveness', is a matter of debate (see *Boulton et al., 2015*). While we view these as overlapping concepts, our findings do highlight reasons for caution in considering them interchangeable. For example, while

the structure of the major axis of **G**, **g**$_{max}$, is broadly consistent with it reflecting an axis of integrated genetic variation in 'stress coping style' (see below), *emergence time*, an OFT trait that is commonly considered to reflect 'boldness' (Burns 2008), was found to be heritable but did not load significantly on **g**$_{max}$ (see *Figure 4*). This could indicate that the shy—bold continuum does not align with the stress coping style continuum in our model organism. However, it could also reflect complexities with interpreting *emergence time*. For example, at least some genotypes (and individuals) could perceive the shelter area as less safe than the open arena (i.e. counter to the assumption that fast emergence reflects a lack of fear of the open arena, and thus greater boldness, e.g. Burns 2008). Indeed, in an earlier study some guppies decreased (rather than increased) shelter use following simulated predation events (*Houslay et al., 2018*).

Tendency to shoal varies among individuals but is not detectably heritable. Although not generally considered a stress-response trait per se, shoaling is an anti-predator behaviour in guppies (*Herbert-Read et al., 2017*). We had therefore predicted that heightened perception of risk in the open field might also be associated with increased shoaling tendency. This was not the case at the among-individual level (*Supplementary file 5*), while the absence of detectable genetic variance meant that we could not test this prediction in **G**.

We find strong evidence of significant additive genetic variance in a key physiological component of the stress response: waterborne cortisol concentrations following exposure of the fish to a handling stressor. Our findings suggest that previously detected differences among individuals in their cortisol response to a stressor (*Houslay et al., 2019*) are primarily attributable to genetic effects, with the estimated heritability ($h^2$ = 0.31) constituting over 80% of the individual–level repeatability ($R$ = 0.37) for ln-transformed *Cortisol*. In addition, by adopting a reaction norm approach to modelling stress physiology, as recently advocated by ourselves (*Houslay et al., 2019*) and others (e.g. *Fürtbauer et al., 2015*; *Hau and Goymann, 2015*; *Taff and Vitousek, 2016*; *Guindre-Parker, 2020*; *Malkoc et al., 2021*), we detect GxE reflecting genetic differences in the extent of habituation to the stressor over repeated exposures. This result is potentially important because poor habituation of the hypothalamic-pituitary-adrenal/interrenal (HPA/I) response to repeated or ongoing stressors can lead to well documented health problems in human and animal populations (*Segerstrom and Miller, 2004*; *Koolhaas, 2008*; *Romero et al., 2009*; *Mason, 2010*). Our detection of heritable variation in the degree of habituation to stressors raises the possibility of developing targeted selection strategies to improve welfare in captive populations (e.g. *Frankham et al., 1986*; *Muir and Craig, 1998*; *Oltenacu and Algers, 2005*).

Our findings also highlight that there is greater additive genetic variance (and heritability) for cortisol levels following the first exposure to the stressor than following the third. This pattern, which occurs because genotypes that produce the highest cortisol response at first exposure also show the most marked habituation, is consistent with the idea of cryptic genetic variance being 'released' by exposure to novel, and so potentially stressful, environments (*Ledón-Rettig et al., 2010*; *Ledón-Rettig et al., 2014*; *Berger et al., 2011*; *Paaby and Rockman, 2014*). All else being equal, it also means that selection on cortisol levels following stressor exposure should induce a stronger evolutionary response in naïve relative to habituated fish. However, the strong positive cross-environment correlation means that the ranking of genotypes with regard to their cortisol responses is consistent across repeated stressor exposures. Thus selection on the (average) GC response would result in a correlated evolutionary response of habituation rate, and vice versa. With regard to drawing inferences from these GC levels about the extent to which an organism is in a 'stressed' state, it is worth noting the potential complexity that may arise from GCs serving numerous functions. For example, in addition to mobilising energy and down-regulating non-essential processes in order to facilitate coping with an acute stressor (*Romero and Beattie, 2022*), GCs such as cortisol are also thought to play a key role in reproductive processes (*Wingfield and Sapolsky, 2003*; *Breuner et al., 2008*; *Bonier et al., 2009b*; *Ouyang et al., 2011*; but see also *Bonier et al., 2009a*). Thus, there is a need for some caution when equating a high GC level with a highly stressed 'state'. Notably, studies of GC levels alone will naturally not capture a complete picture of among-individual variation in stress physiology. The HPA/I axis (which culminates in GC secretion) reflects just one of several important neuro-endocrine stress response pathways (see *MacDougall-Shackleton et al., 2019*) whose wider integration, while logistically challenging to study at present, strongly merits attention going forward (*Gormally et al., 2020*).

Considering all traits together, **G** shows evidence of genetic integration between behavioural and endocrine components of the stress response. The ends of the major axis of **G** ($g_{max}$) are largely consistent with expectations given genetic variance in 'stress coping styles' (**Koolhaas et al., 1999**). Genotypes showing (putatively) more proactive 'flight' type behaviour in the OFT (i.e. thigmotaxis, high *track length*, low *relative area covered,* few *freezings*) produce higher levels of cortisol following the handling and confinement stressor, while genotypes showing (putatively) more reactive 'freezing' type behaviour in the OFT (i.e. low *track length* and many *freezings*, in addition to higher *relative area covered* and more *time in the middle*) produce lower levels of cortisol following the same stressor. Were one to accept this view, the trait loadings on $g_{max}$ would suggest that thigmotaxis and freezing behaviour are strong indicators of stress coping style. Notably, here we find continuous variation along this major axis of genetic variation (see *Figure 4—figure supplement 2*), rather than the bimodal distribution suggested by some of the coping styles literature (**Koolhaas et al., 1999**; **Koolhaas et al., 2007**; **Koolhaas et al., 2010**). An alternative interpretation, however, is that $\mathbf{g_{max}}$ represents variance in stress response 'magnitude' rather than coping style per se. This is because the putatively 'reactive' end of $\mathbf{g_{max}}$ might simply reflect a low magnitude stress response rather than a particular style of 'coping' with a stressor. Such an interpretation would help explain why fish (or genotypes) that produce lower levels of cortisol following handling and confinement also have more 'exploratory' movement (higher *relative area covered*) and reduced thigmotaxis (i.e. increased *time in the middle*) in the OFT. We note that the distinction between 'style' and 'responsiveness' may be rather moot if, for example, fish with proactive styles are also more responsive.

In fact, a subsequent 'two-tier' iteration of the coping style model proposed that variation in the magnitude of the stress response (termed 'stress reactivity') could be viewed as a second dimension of variation, distinct from differences in the 'type' (or style) of response (**Koolhaas et al., 2010**). It is not clear to us that these two dimensions, if both present, can be tested for and disentangled empirically in the current data—for example, there is no a priori expectation that style and magnitude should manifest as orthogonal axes in **G** among traits analysed here (and so both could conceivably align on $\mathbf{g_{max}}$). In principle, disentangling these dimensions should be possible by characterising stress response through within-subject comparison of behaviour and endocrine state between stressed and 'baseline' (unstressed) states. In practice, we found here that genetic variation in GC responsiveness (i.e. the slope across states, accepting that habituation represents a form of this) is very strongly correlated with the average (or intercept). We found similar results for behavioural responses to increased stressor severity in another study of this population, where there exist strong among-individual correlations between the intercept and slope of the plastic response (**Houslay et al., 2018**). While we cannot claim a true 'baseline' state in either case, these results suggest that stress coping style and the magnitude of response do both exist as axes of variation, but may also be tightly correlated.

The genetic integration of behaviour and physiology detected here is consistent with (but not proof of) the hypothesis that correlational selection in the past has led to the coevolution of these components of the stress response. This hypothesis assumes that correlational selection favours any combination of trait values that yield higher fitness, creating a genetic correlation as mutations that generate such combinations will be recruited into the population while those that do not will be selected out (**Lande, 1980**; **Roff and Fairbairn, 2012**). While reflections on the role of past correlational selection are necessarily speculative, we can say that the structure of **G** should shape (and potentially constrain) future evolutionary responses to selection—whether natural or artificial. Here we have no direct knowledge of how contemporary selection is acting in the wild or whether it might be changing as a consequence of anthropogenic stressors. Nor do we know exactly how well our estimate of **G** will match that which may be found in the wild (although our study animals are all recent descendants of wild-collected animals). Thus, we cannot comment directly on how **G** will shape future evolution of the guppy stress response beyond noting that selection on behaviour will cause correlated evolution of endocrine physiology (and vice versa). Nonetheless, while it seems reasonable to expect that current integration of stress response in natural populations should be broadly adaptive, this seems less likely in captive populations (at least for species without a long history of domestication and opportunity for adaptation to artificial environments). We know that prolonged, chronic activation of stress response pathways (notably the HPA/I axis) frequently disrupts health and survival in captive animals (**Huether, 1996**; **Boonstra and Fox, 2013**). It may be that more stress-responsive genotypes are disadvantaged in novel artificial conditions (e.g., if acute stress responsiveness positively predicts susceptibility to

chronic stress). However, even if true this would not imply high (acute) stress-responsiveness was also disadvantageous in the wild. Since natural selection should purge alleles that are universally detrimental, it seems more plausible that genetic variation along the major axis described here is maintained by some form of selective trade-off (as widely hypothesised for maintenance of personality variation; e.g., *Stamps, 2007*; *Wolf et al., 2007*; *Réale et al., 2010*). For instance, genotypes susceptible to harm under chronic stressor exposure will likely persist in populations if they also confer advantages under an acute stress challenge. In natural populations not only is exposure to acute stressors more common than to chronic stressors, but also selection through chronic stress exposure may be conditional on (and subsequent to) surviving acute challenges (such as predator attacks).

## Conclusions and future Directions

Here, we find evidence for genetic variation in – and integration of – behavioural and physiological (endocrine) components of the stress response. Overall, we consider the structure of **G** to be broadly consistent with the widely invoked 'reactive—proactive' model of variation in stress coping style (*Koolhaas et al., 2007*). This interpretation rests largely on the structure of behavioural variation revealed by the OFT, which is dominated by a major axis running from genotypes with more proactive 'flight' type stress responses to those with more reactive 'freeze' type stress responses. Endocrine traits align with this axis: genotypes exhibiting 'flight' type responses show higher cortisol levels (and exhibit faster habituation of GC physiology) when subjected to repeated handling and confinement stressors than those exhibiting more 'freeze' type responses. However, as these latter genotypes with more 'freeze' type responses and low cortisol responses to stressors also tend – on average – to display space use patterns characteristic of exploration and reduced thigmotaxis (potentially indicative of being relatively unstressed), an alternative interpretation is that $g_{max}$ primarily reflects differences in stress responsiveness. Further distinguishing between 'style' and 'responsiveness' may depend on their association, for example if fish with proactive styles are also more responsive. Although future studies could certainly target separation of these dimensions further, we think greater insights may come from expanding the set of traits (in particular, to include other components of the physiological stress response, as noted above) and/or stress contexts (i.e. exposing the subjects to different types of stressor). With respect to the latter, here we observed behavioural and GC responses to two different stressors separated in time. We think that our demonstration of genetic correlation structure between behaviours tested in one context and physiology assayed in a different one adds at least some support to the idea that the integration characterised here may ultimately prove generalisable across stress contexts.

Our results suggest that continued evolution of stress-related behaviour will have consequences for glucocorticoid physiology and *vice versa*. Determining contemporary selection on the stress response, and testing the possibility that its underlying genetic variation is maintained by fitness trade-offs, is thus an obvious – if empirically challenging – next step to understanding the functional importance of genetic variation in the stress response in wild populations. In a more applied context, integration of behavioural and endocrine stress response components at the genetic level has potential utility for artificial selection to improve resilience to chronic stressors in managed populations (*Gebauer et al., 2021*). Specifically, it may be possible to identify non-invasive, high throughput, behavioural biomarkers and target them in selection schemes to reduce chronic activation of the HPA/I endocrine axis and its attendant deleterious effects.

## Materials and methods
### Husbandry and breeding

We used fish taken from our captive population housed at the University of Exeter's Penryn campus, which is descended from wild fish collected in 2008 from the lower Aripo River in Trinidad. This population has been maintained at a population size of several thousand and has undergone no deliberate selection or inbreeding. All fish are fed to satiation twice daily (0800–1000 hr and again at 1600–1800 hr) using commercial flake food and live *Artemia nauplii*. Water temperature is maintained at 23–24°C in well-aerated closed system tank stacks that undergo 25% water changes each week and with weekly tests for ammonia, nitrate and nitrite levels. Lighting is kept at a 12:12 light/dark cycle. The experiment described in this study was carried out in accordance with the UK Animals (Scientific

Procedures) Act 1986 under licence from the Home Office (UK), and with local ethical approval from the University of Exeter.

To create our pedigreed sub-population, female fish were sampled haphazardly from the stock tanks and kept in female-only groups for 3 months. Isolation from male contact minimised the chance of females carrying viable sperm from previous matings. For the first generation of offspring, we used a group breeding design (as detailed in *White et al., 2018*); briefly, females were tagged under anaesthetic (buffered MS222 solution) using visible implant elastomer (VIE) to allow individual identification. We then assigned groups of four females to one male in 15 L breeding tanks (18.5cm x 37 cm x 22 cm), and inspected females daily for high gravidity (swollen abdomens and enlarged 'gravid spots'). Heavily gravid females were then isolated in 2.8 L brood tanks to give birth (and were returned to the breeding tanks either after producing a brood or two weeks of isolation). Any offspring produced in the breeding tanks were excluded from the experiment as maternal identity could not be positively identified. For the following generations, after 3 months of isolation from males we moved females into individual 2.8 L tanks, with one male then circulated among three females. Males were moved between females every 5–8 days. In this way, females did not have to be moved to brood tanks, and any offspring could be assigned to mothers definitively. In this setup, offspring were moved to a separate brood tank on the day of birth. Note that as the gestation period for guppies is approximately 1 month, any brood produced by a female less than one month after exposure to their designated male was recorded in the pedigree as having unknown paternity.

Within 24 hr of a female producing a brood, we recorded her weight (g) and brood size. We kept juvenile fish in full-sib family groups in 2.8 L tanks before moving them to 15 L 'growth' tanks at an average age of 56 days. At an average age of 133 days (range 59–268), we tagged individuals and placed them into mixed family groups of 16–20 adults (with an even mix of males and females), kept in 15 L tanks. Note that variation in tagging age arose largely because groups were necessarily established sequentially as sufficient individuals from multiple families reached a large enough size that we deemed the procedure to be safe. Each adult group comprised a mix of fish from different families, reducing the potential for common environment effects to upwardly bias our genetic parameter estimation.

## Overview of behavioural phenotyping

Behavioural phenotyping commenced at least one week after tagging. In all trials, we filmed movement behaviour of individual fish using a Sunkwang video camera equipped with a 6–60 mm manual focus lens suspended over the tank. We used the tracking software Viewer II (BiObserve) to extract behavioural data from each recording (detailed below). The tank was lit from below using a light box and screened with a cardboard casing to prevent external visual disturbance. After each behavioural trial, the individual tested was weighed and then moved to a temporary 'holding tank'. Once a full group (as described above) had been tested, all were moved from the holding tank back to their home tank. We replaced the water in the testing and holding tanks between groups to reduce the build-up of hormones or other chemicals. The first offspring generation experienced four repeat open field trials (OFTs) over a 2-week period, with at least 48 hr between trials. Subsequent generations experienced four repeat behavioural trials, alternating 2 OFTs with two emergence trials (ETs). For the final two generations, we extended the OFTs by including a shoaling trial (ST) at the end of each OFT.

*Open field trials (OFT)* followed the methodology described by *White et al., 2016*. Briefly, we assessed individual behaviour in a 20cm x 30 cm tank, filled to a depth of 5 cm with room-temperature water from the main supply. We caught fish individually from their home tank, examined them quickly for identification tags, then placed them immediately into the centre of the OFT tank. After allowing 30 s for acclimation, we filmed behaviour for 4m30s. Behaviours characterised from the tracking software were *track length* (the total distance the fish moved during the trial; cm), *area covered* (the percentage of 1cm x 1 cm grid squares through which the fish moved during the trial; %), *time in middle* (time spent in a rectangular inner zone which was defined as being the same size as an outer area; seconds), and number of *freezings* (defined in practice as the number of times that an individual's velocity dropped below 4 cm/s for a minimum of 2.5 s; *White et al., 2018*).

*Shoaling trials (ST)* were appended to a subset of OFTs, by positioning a small tank containing 10 stock fish (of same sex as the test subject) next to one end of the OFT tank but with visual cues blocked by a cardboard divider. At the end of the normal OFT, we removed this divider slowly, allowing the

focal animal to have visual contact with the shoal. We began recording the shoaling trial 30 s after removing the divider in order to limit any artefacts of slight disturbance. (Note that we used a further cardboard casing around the shoaling tank to avoid any additional external visual stimulus). We then recorded behaviour of the test fish for an additional 3 min. We characterised *shoaling tendency* via the tracking software by subdividing the tank area into three equal-sized rectangular areas: one next to the tank holding the group of same-sex fish, one farthest from this group, and the central area. We then calculated *shoaling tendency* as the time spent in the 1/3 area closest to the same-sex group after subtracting the time spent in the 1/3 area farthest away. The decision to use a single-sex shoal aimed to reduce any effects of (potential) mate preference and/or avoidance, but also this necessitated replicate arena setups allowing male and female individuals from each group to be tested in the OFT/ST concurrently. We randomised which tank was used for each sex in each group and recorded this information.

*Emergence trials (ET)* followed the methodology described by *White et al., 2016*. Briefly, we tested individuals in a 20cm x 40cm tank, filled to a depth of 8 cm with room-temperature water from the main supply. A 10 cm section of the tank length was walled off creating a shelter area (20cm x10cm), the walls and floor of which were painted black. The focal fish was placed into the shelter area and allowed to acclimate for 30 s, at which point we opened a sliding door to allow access to the rest of the tank, which was brightly lit from below and otherwise bare. *Time to emerge* (in seconds) was recorded by the tracking software automatically as the fish exited the shelter area and emerged into the open tank section. Trials were ended either at emergence or at 15 min if the fish had not emerged by that point; in the case of non-emergence, fish were given the maximum value (i.e. 900 s).

## Derivation of 'relative area' from OFT trials

The '*area covered*' variable assayed in the OFT is calculated in BiObserve by dividing the arena (i.e. the total area of the tank as viewed from the camera) into 1cm x 1cm grid squares. The path taken by the fish during observation is then used to determine what proportion of these grid squares the fish entered. However, we sought to derive a measure of '*relative area*' that describes whether a fish covers a large, or small area relative to its observed *track length*.

To do this, we simulated 'random swims' within the arena across the observed range of *track lengths*. We first selected 40 OFT results at random from our total data set and extracted the coordinates of the fish in each frame from the raw tracking file, creating a set of x and y movements and their associated distances. As original coordinates were recorded in pixels we used the calibration of the software to convert to cm units. We then use a 'random walk' algorithm to select a movement (i.e. step size and direction) from this observed distribution at random and calculate the new coordinates. If the movement keeps the 'fish' within the bounds of the 'tank' (i.e. defined as a 20cm x 30 cm arena), the movement is accepted and coordinates added to a movement matrix; if not, a new movement is drawn from the distribution. If the movement is greater than 1 cm in distance, we break the movement into a number of smaller parts to be added to the matrix (such that we capture the coordinates of grid squares through which the 'fish' moved along the way). Once the total distance of the random walk reached or exceeded the *track length* set as the simulation objective, the path is terminated and the area covered is calculated by counting the number of unique grid squares in the matrix of coordinates and dividing by the total number possible.

After simulating random walks across 500 values of *track length* (using a vector of 100 values evenly spaced across the range of true data, repeated five times), we modelled (simulated) area covered as a fourth order polynomial function of *track length*. Using this regression model (which explained 97.8% of the variance in simulated data), we calculated the *relative area* for each actual OFT trial as the observed area covered minus the area covered under a random swim, as predicted from our regression model and the observed *track length*.

## Waterborne hormone sampling

On completion of behavioural data collection, individuals entering the endocrine testing program were left undisturbed for a minimum of two weeks. Waterborne hormone sampling was then conducted over a 5-day period that included three handling and confinement stressor exposures with 48 hr between each. We followed the method described by *Houslay et al., 2019* to obtain repeated non-invasive GC measures of individuals using holding water samples from the first and third confinements. Note

that only two samples per fish were analysed because the financial and time costs of analysing three was deemed prohibitive. We nonetheless applied the stressor stimulus three times as our prior study showed this was sufficient to produce a strong habituation response, that is, a significant decrease in water-borne cortisol over the three sampling periods (*Houslay et al., 2019*).

We collected samples between 1200 and 1400 hr to control for diel fluctuations in GC levels. For each sample, we netted an entire group from their home tank quickly using a large net, transferring them to two holding tanks (containing water from the home tank supply) for moving to an adjacent quiet room (performed within 20 s of the net first hitting the water). We then transferred fish to individual Pyrex beakers containing 300 ml of clean water from the main supply (which serves the main housing units), which had been warmed to the appropriate temperature (mean = 24.1 °C, range 23–24.9°C). Beakers were placed within cardboard 'chambers' to prevent fish from seeing each other or experiencing outside disturbance. One fish was transferred every 30 s, alternating across holding tanks, such that all fish were in their beakers within 10 min of the initial netting. After 60 min in the beaker, each fish was removed by pouring its sample through a clean net into a second beaker, with the fish then quickly checked to confirm ID and returned to the holding tank until the entire group could be returned to its home tank.

We immediately filtered each water sample using Grade one filter paper (Whatman), then passed them slowly through solid phase C18 extraction columns (Sep-Pak C18, 3 cc, Waters) via high-purity tubing (Tygon 2474, Saint Gobain) under vacuum pressure (*Earley et al., 2006*). Columns were primed beforehand with 2 × 2 ml HPLC-grade methanol followed by 2 × 2 ml distilled water and were washed afterwards with a further 2 × 2 ml distilled water to purge salts. We then covered both ends of each column with film (Parafilm M, Bemis) and stored them at –20°C for future analysis. We washed all beakers, tubes and funnels with 99% ethanol and rinsed them with distilled water prior to each sampling procedure. The remainder of the endocrine assay procedure involved elution, resuspension, separation, and quantification of free cortisol by enzyme immunoassay (EIA) using Cayman Chemicals, Inc EIA kits. Detailed methods are exactly as described by *Houslay et al., 2019* and so not repeated here (note that here we assayed the free fraction of cortisol only). To validate the cortisol kits, we examined whether the kit standard curve was parallel to a serial dilution curve derived from pooled guppy water-borne hormone extract. Twenty µl was taken from each of the male samples and pooled; 20 µl was taken from each of the female samples and combined into a separate pool. A total of 400 µl of the pools was serially diluted from 1:1 to 1:128 and these samples were assayed alongside the kit standard curve on two occasions (June and December 2017, marking the start and finish of sample processing). All dilution curves were parallel to the standard curve (slope comparison test, *Zar, 1996*, p.355; June, male: $t_{12}$ = 0.029, p = 0.97; June, female:: $t_{12}$ = 0.343, p = 0.74; December, male:: $t_{12}$ = 0.119, p = 0.91; December, female:: $t_{12}$ = 0.224, p = 0.83). The serial dilution curves also identified 1:32 as an appropriate dilution to ensure that all samples fell on the linear phase of the standard curve. A total of 37 96-well plates were used, and the pooled sample was included at the beginning and end of each plate. Intra-assay coefficients of variation ranged from 0.12% to 19.83% with a median of 3.08%; the inter-assay coefficient of variation was 19.22%. Cortisol is presented and modelled in (ln-transformed) units of ng/hr to reflect the 1 hr sampling duration.

## Statistical methods

All data handling and analysis was performed in R version 3.6.3 (*R Development Core Team, 2020*). We used the 'tidyverse' set of packages for data handling and visualisation (*Wickham, 2017*), and ASreml-R v4 (*Butler, 2021*) for fitting linear mixed effects models (as described in full below). We also used 'nadiv' for pedigree preparation (*Wolak, 2012*). All models fitted assumed (multivariate) Gaussian error structures, and we visually assessed residuals to verify this was reasonable (after data transformation in some cases). To test for significance of among individual and/or genetic (co)variance components, we fitted nested models with different random effects structures and compared them using likelihood ratio tests (LRTs). We calculated $\chi^2_{nDF}$ as twice the difference in model log likelihoods, with the number of degrees of freedom (*n*) equivalent to the number of additional parameters in the more complex model. When testing a single random effect (variance component), we assumed the difference to be asymptotically distributed as an equal mix of $\chi^2_0$ and $\chi^2_1$ (denoted $\chi^2_{0,1}$; *Self and Liang, 1987*; *Visscher, 2006*).

For each OFT, ST, and ET behaviour in turn (*relative area, time in middle, track length, freezings, shoaling tendency,* and *emergence time*), we used the random effects specification to partition phenotypic variation ($V_p$, conditional on fixed effects as described below) into the effects of additive genetics ($V_a$), permanent environment defined as the non-(additive) genetic component of among-individual differences, ($V_{pe}$), and housing group ($V_{group}$), as well as residual variation ($V_{residual}$). We natural log-transformed *emergence time* prior to analysis to meet assumptions of residual normality and homoscedasticity. For all behavioural traits, we included fixed effects of assay *repeat*, the *order within each group* in which the fish was trialled (mean-centred continuous predictor), *temperature* (mean-centred and scaled to standard deviation units), *time* (in minutes from midnight, mean-centred and scaled to standard deviation units), *age* (mean-centred and scaled to standard deviation units), *sex*, and the *generation* from the breeding population. The *order caught* predictor in particular was used to control statistically for variation in disturbance over the course of measuring a group (*White et al., 2018*). For *shoaling tendency* only, we incorporated an additional fixed effect of *setup* (as detailed above). We tested the significance of genetic variance for each behaviour by LRT comparison of the corresponding full model to one in which the (additive) genetic random effect was excluded.

Cortisol data were also natural log (ln) transformed for analysis. We formulated a bivariate model to test for both additive genetic variation and genotype-by-environment interaction (GxE) in cortisol levels across the two 'contexts' (i.e. samples retained for each individual at first and third confinement, denoted Cortisol$_1$, Cortisol$_3$). Random effects were first used to partition phenotypic (co)variance (conditional on fixed effects) into among-group and residual components. Fixed effects included the context-specific means, and overall effects of the *order* in which the fish was caught from each group for assay (mean-centred continuous predictor), *temperature* (mean-centred and scaled to standard deviation units), *time of day* (mean-centred and scaled to standard deviation units), *age* (mean-centred and scaled to standard deviation units), and *sex*. In addition, we included fixed covariates of *body mass* (mean-centred and scaled to standard deviation units) and a *sex* by *body mass* interaction (see *Houslay et al., 2019* for rationale of controlling for body size effects on waterborne hormone levels in this way). Note that modelled in this way each individual is sampled only once for each context-specific cortisol trait so no random effect of individual identity is included. To test for additive genetic variation ($V_a$) we compared this first bivariate model to a second formulation that also included the (additive) genetic merit, but under the assumption that this is invariant with context within an individual (such that $V_{a1} = Va_{a3}$ and $r_{a1,3} = 1$ and there is no GxE). We then test for the GxE by comparing the second model to a third in which we allow GxE (i.e. the context-specific genetic variances are free to differ and the cross-context genetic correlation can be <+1).

Lastly, we built a multivariate animal model to estimate **G** and to test the hypothesised genetic integration among behavioural and physiological stress components. We retained only response traits that harboured significant $V_a$ as shown in univariate models, and so the final model comprised response traits *relative area, time in middle, track length, freezings* (square root transformed), *emergence time* (ln transformed), and *Cortisol* (ln transformed). We multiplied (transformed) *emergence time* by –1 to simplify interpretation (higher values represent faster emergence). We also scaled all (transformed) response variables to standard deviation units. This was to facilitate model fitting, and also prevent scale effects complicating interpretation of eigenvectors of **G**. Fixed and random effects were fitted on each trait as specified for the univariate models. Note that one exception to this is that we elected to treat *Cortisol* as a single repeated-measures trait here (with two repeats, one per context) such that a permanent environment effect was now included. Fixed effects estimates are reported in the supplementary information (*Supplementary file 1*).

We specified additive genetic (**G**), permanent environment (**PE**), group (**GROUP**), and residual (**R**) covariance structures as unstructured matrices to be estimated. Note that **R** partitions observation-level covariances (as opposed to individual-level in **PE**) that are not definable or statistically identifiable if traits are not measured at the same time (i.e., all covariances relating to *emergence time* or *Cortisol*). Where this was the case we constrained specific covariance terms in **R** to equal zero. Estimates of **PE**, **GROUP** and **R** are provided in *Supplementary files 2-4*. We tested for overall additive genetic covariance among the traits by comparing this model against a reduced one in which **G** was specified as a diagonal matrix (i.e., additive genetic variances are estimated but covariances are assumed to equal zero). To aid the interpretation of covariance terms contained in **G,** we calculated the corresponding genetic correlations $r_a$ from the full model. For any pair of traits (x,y), $r_{a(x,y)} = COVa_{a(x,y)}/ (\sqrt{(V_{a(x)})} \times \sqrt{(V_{a(y)})})$.

We also subjected our estimate of **G** to eigen decomposition to determine the proportion of additive genetic variation captured by each principal component and assess whether the major axis of variation ($\mathbf{g}_{max}$) could indeed explain most of the genetic variance in the multivariate phenotype. We estimated uncertainty on the trait loadings associated with each principal component (eigenvector), as well as on the variance explained by each eigenvector, using a parametric bootstrap approach as described by *Boulton et al., 2014*.

For visualisation of bivariate relationships at the additive genetic level, we used the R package 'ellipse' (*Murdoch and Chow, 2018*) to determine the coordinates of an ellipse representing the approximate 95% confidence region of deviations based on the point estimate of **G**. We repeated this procedure for the corresponding regions defined from 5,000 bootstrapped values of **G** (i.e. to indicate uncertainty arising from estimation of the genetic covariance structure itself). Best linear unbiased predictors (BLUPs) are used for visualisation only, not for any statistical analysis (*Houslay and Wilson, 2017*).

To test for associations between all traits (i.e., including *shoaling tendency*) at the among-individual level, we also built a multivariate model as above with the addition of *shoaling tendency* and without estimating additive genetic effects. The estimates of all among-individual (co)variances are provided in *Supplementary file 5*.

## Acknowledgements

We are grateful to P Sharman, D Maskrey and C Mnatzagian for help in the lab, and to J Martin, J Styga, P Vullioud and all members of the Wilson group for discussion. We also thank J McGlothlin and two anonymous reviewers for comments and suggestions that greatly improved this manuscript.

## Additional information

### Funding

| Funder | Grant reference number | Author |
| --- | --- | --- |
| Biotechnology and Biological Sciences Research Council | BB/L022656/1 | Ryan L Earley Andrew J Young Alastair Wilson |
| Biotechnology and Biological Sciences Research Council | BB/ M025799/1 | Ryan L Earley Andrew J Young Alastair Wilson |

The funders had no role in study design, data collection and interpretation, or the decision to submit the work for publication.

### Author contributions

Thomas M Houslay, Conceptualization, Data curation, Formal analysis, Investigation, Methodology, Project administration, Validation, Visualization, Writing - original draft, Writing – review and editing; Ryan L Earley, Conceptualization, Formal analysis, Funding acquisition, Investigation, Methodology, Project administration, Resources, Supervision, Validation, Writing – review and editing; Stephen J White, Data curation, Investigation, Project administration, Writing – review and editing; Wiebke Lammers, Andrew J Grimmer, Investigation, Project administration, Writing – review and editing; Laura M Travers, Elizabeth L Johnson, Investigation, Writing – review and editing; Andrew J Young, Conceptualization, Funding acquisition, Supervision, Writing – review and editing; Alastair Wilson, Conceptualization, Funding acquisition, Methodology, Project administration, Resources, Supervision, Writing - original draft, Writing – review and editing

### Author ORCIDs

Thomas M Houslay  http://orcid.org/0000-0001-5592-9034
Stephen J White  http://orcid.org/0000-0002-8538-6066
Elizabeth L Johnson  http://orcid.org/0000-0002-2009-7685

## Ethics

The experiment described here was carried out in accordance with the UK Animals (Scientific Procedures) Act 1986 under licence from the Home Office (UK), and with local ethical approval from the University of Exeter. Individual tagging for identification purposes was performed via injection of coloured elastomer under MS222 sedation, and every effort was made to minimise suffering.

## Decision letter and Author response

Decision letter https://doi.org/10.7554/eLife.67126.sa1
Author response https://doi.org/10.7554/eLife.67126.sa2

## Additional files

### Supplementary files

• Supplementary file 1. Fixed effects estimates from the full multivariate animal model.

• Supplementary file 2. Permanent environment (co)variance matrix from the full multivariate animal model.

• Supplementary file 3. Group (co)variance matrix from the full multivariate animal model.

• Supplementary file 4. Residual variance-correlation matrix from the full multivariate animal model.

• Supplementary file 5. Among-individual (co)variance matrix from the multivariate model that excluded genetic effects.

• Supplementary file 6. Eigen decomposition of the G matrix.

• Transparent reporting form

### Data availability

Data and analysis code have been deposited in Dryad.

The following dataset was generated:

| Author(s) | Year | Dataset title | Dataset URL | Database and Identifier |
|---|---|---|---|---|
| Houslay TM, Earley RL, White SJ, Lammers W, Grimmer AJ, Travers LM, Johnson EL, Young AJ, Wilson AJ | 2021 | Genetic integration of behavioural and endocrine components of the stress response | http://dx.doi.org/10.5061/dryad.z34tmpgcg | Dryad Digital Repository, 10.5061/dryad.z34tmpgcg |

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

## Appendix 1

### Selection of OFT behaviours

Behavioural traits from the Open Field Trials (OFTs) were selected based upon previous research in this population and typical OFT measurements that describe variation in movement behaviour. Here we briefly describe selection of some traits against potential alternatives.

#### i. 'Freezing' behaviour

Freezing behaviour is important to the 'coping styles' model as this is often used to describe the reactive style of coping. A velocity threshold for active swimming (4 cm/s) has been defined for this population and used in previous studies for percentage of time spent active and for the number of freezings (e.g., *White et al., 2018*). We assessed 'inactivity' (i.e., 100 – time spent active, the percentage of time spent below the velocity threshold), but this was strongly correlated with 'track length' at the observation level ($r$ = –0.96, 95% CI = [-0.97,–0.96], t = –210.3, $P$ < 0.001; *Appendix 1—figure 1*) and so was considered not to add any further value. We also fit a bivariate animal model (with fixed and random effects as in the main text), and found that all correlations were >0.86, including a genetic correlation of $r_A$ = 0.99 ± 0.01. We elected to retain track length rather than inactivity because it is a raw value rather than percentage, and so provides useful information on (for example) extreme values of track length.

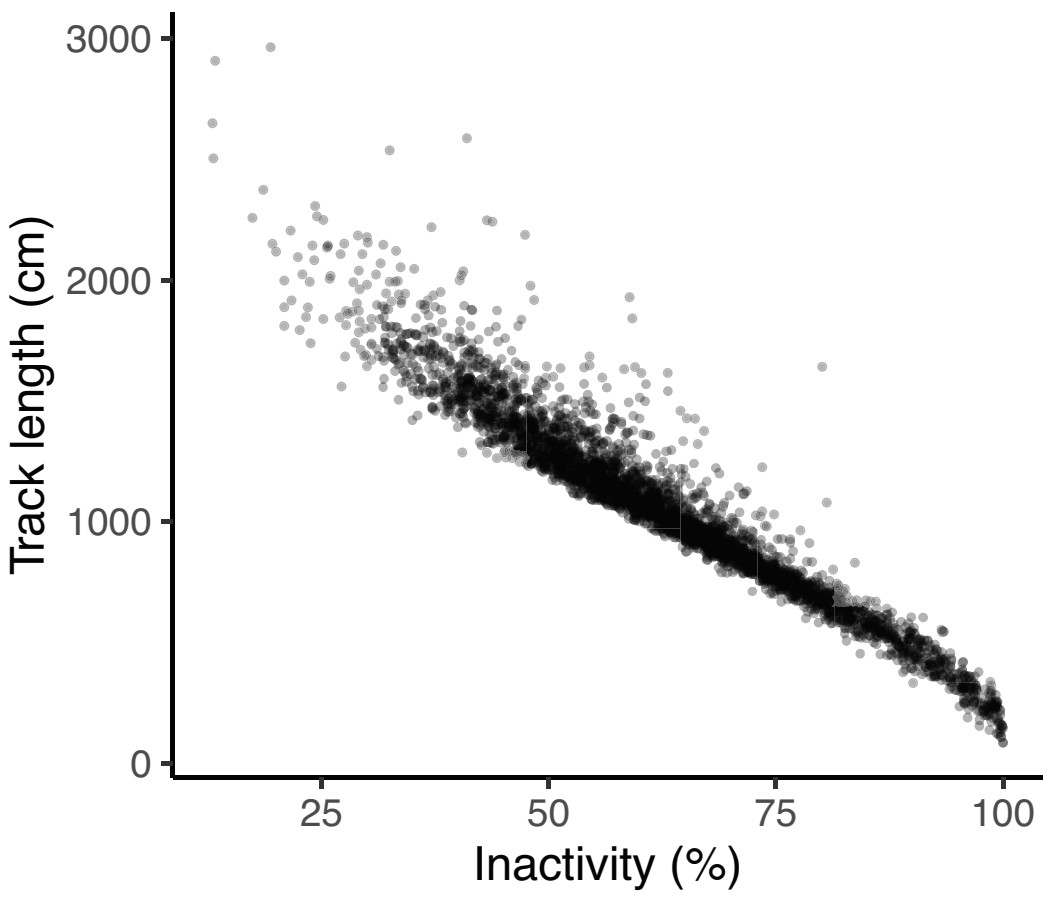

**Appendix 1—figure 1.** Percentage of time inactive is strongly correlated with track length at the phenotypic level.

#### ii. Thigmotaxis

OFTs are often used for research in personality traits (such as 'boldness' or 'exploration') as well as for anxiety behaviours. In the former, it is typical to define a central zone and quantify

the time spent exploring this region as a measure of boldness. In the latter, it is typical to consider the average distance from the arena wall. Here we found that these measurements are highly correlated at the observation level ($r = 0.94$, 95% CI = [0.94,0.94], $t = 159.2$, $P < 0.001$; *Appendix 1—figure 2*). We also fit a bivariate animal model (with fixed and random effects as in the main text), and found that all correlations were >0.9, including a genetic correlation of $r_A = 0.99 \pm 0.01$. These results suggested no gain to interpretation of using both, and we elected to retain time in the middle as the zoning approach is standard in much animal personality work.

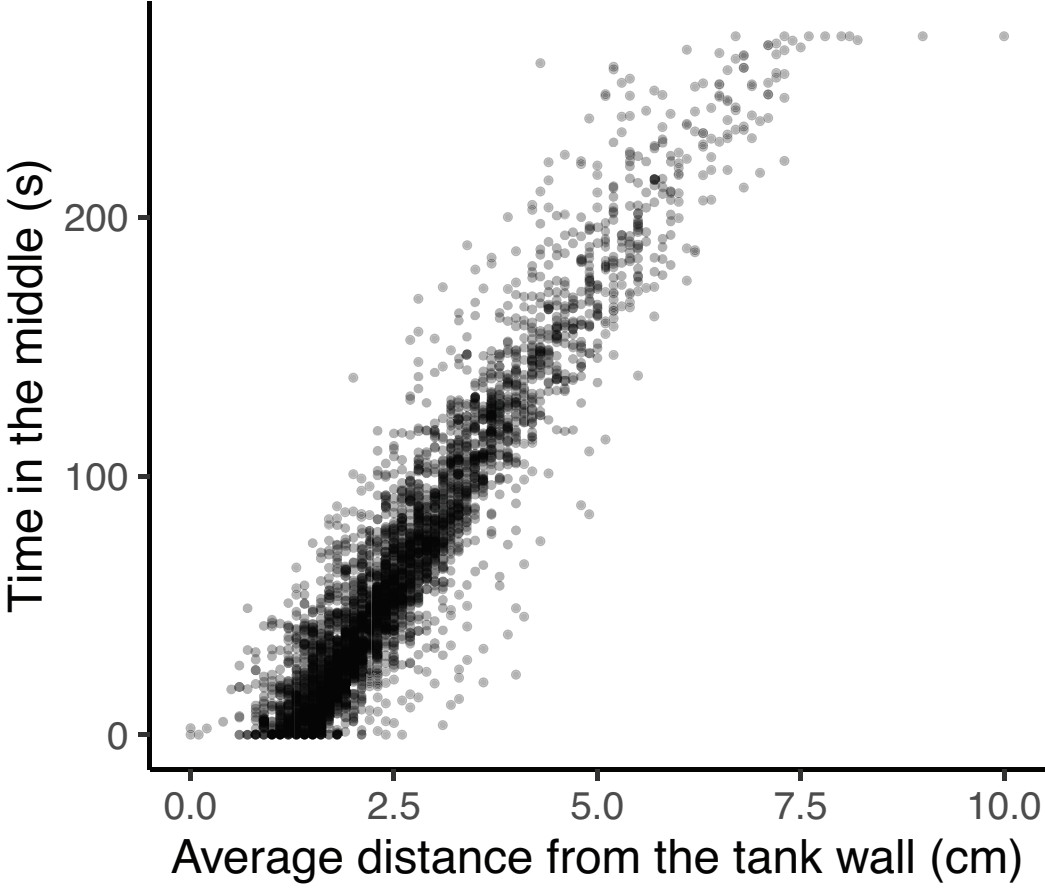

**Appendix 1—figure 2.** Time in the middle is strongly correlated with average distance from the tank wall at the phenotypic level.

