## [Editor Report]

This is a timely paper on the genetic integration of behavioral and physiological components of the stress response in guppies. Using evolutionary quantitative genetic approaches, the authors show that genetic variation in the cortisol stress response is associated with genetic variation in stress-related behaviors. This result suggests that physiological and behavioral responses to stress should show correlated evolution in response to natural selection, which is of interest to evolutionary biologists and for animal welfare. The revised manuscript fully addresses both conceptual and methodological limitations of the earlier submission. Congratulations on a nice contribution to the literature.

---

## [Decision Letter]

**Decision letter after peer review:**

Thank you for submitting your article "Evolutionary genetic integration of behavioural and endocrine components of the stress response" for consideration by *eLife*. Your article has been reviewed by 3 peer reviewers, and the evaluation has been overseen by a Reviewing Editor and Molly Przeworski as the Senior Editor. The following individual involved in review of your submission has agreed to reveal their identity: Joel McGlothlin (Reviewer #1).

The reviewers have discussed their reviews with one another, and the Reviewing Editor has drafted this summary to help you prepare a revised submission.

Essential revisions:

1. Extract measures of passive (freezing) versus active (flight/escape) behavioral responses from the video recordings and include them in the analyses of heritability and genetic covariance. Evaluate whether these alternative behavioral measures provide improved discriminatory power between a "reactive-proactive" model versus continuous variation in stress responsiveness.

2. Clearly state the criteria that define a "single major axis of genetic variation," or revise to avoid this strong claim.

3. Provide a complete table of all the eigenvectors and eigenvalues for the eigendecomposition analysis of G.

4. Address the potential effects of using non-standardized stressors across phenotypes on variance component estimation (especially between the approach used to measure GCs versus behaviors).

*Reviewer #1 (Recommendations for the authors):*

Overall, I thought this paper was fantastic. My public review reflects my enthusiasm for the work. Most of my points below amount to writing suggestions.

My only substantive comment is that I think it would be helpful to show a table of all the eigenvectors and eigenvalues, not just the figure showing the first eigenvector loadings. This could be put into the supplement with the other tables.

70-71: Need some more citations here: Maybe Cheverud 1982 (which is cited later), Roff and Fairbairn 2012

143: This is just a style preference, but I always prefer past tense when it comes to the word "predict" in the introduction. By the time the paper is being written, the prediction is in the past. There are some places in the results like this where it is written in present tense but seems like it should be in past tense.

146-147: This sentence is perhaps too informal.

162: Multivariate doesn't need to be in parentheses.

314-327: The organization of this paragraph could be improved a bit. It was tough to follow the results and I had to read it over again to get it.

Figure 3: Would these make more sense with cortisol on the X? I know it's not a regression, but people tend to think of the path hormone -> behavior.

479: I think it might be useful to have one more logical step linking the results to potential past correlational selection. Specifically, some explanation about how these traits might interact to influence fitness would help. It's not entirely clear whether the authors are envisioning correlational selection on a suite of behaviors or correlational selection on hormone levels plus one or more of the behaviors. I'm obviously sympathetic to the idea that correlational selection influences hormone-mediated suites, but I do worry that others might see discussing of it here as too speculative (see below). The authors don't necessarily need to talk about it to explain the significance of their results.

507: I may be misunderstanding this, but wouldn't the reactive-proactive model be consistent with a single axis as well? Reactive would be on one end of the spectrum and proactive would be on the other. This goes back to lines 150-159. The way it is described sounds like a single axis. Is the difference just that the two ends of the continuum are more extreme in the reactive-proactive model?

514: Following on the comment above, I like the idea of correlational selection explaining the effect, but it might be too much of a speculation to lead off the last paragraph of the paper with it. In other words, I think the results are consistent with it, but don't quite suggest it.

*Reviewer #3 (Recommendations for the authors):*

Title: I am not sure what "evolutionary genetic integration" means. "(Quantitative) Genetic integration" sure, but what does the "evolutionary" add? All biological states are the product of evolution, so in a sense "evolutionary" could apply to all biological objects but be quite pointless. The study describes the current state of G in the population but does not describe an evolutionary process (past or on-going).

L.69-73 I agree that past correlational selection would have tended to shape the G matrix along a major axis, but it is difficult to predict a priori what a G matrix should look like if you do not know the variance-covariance of mutational (and migration) input was or what were the exact properties of correlational selection. So "should" we really expect genetic and phenotypic integration? The expectation is not clear to me. Maybe make clearer what the expectation assumes.

L.72 First occurrence of the word "integration" in the main text. Maybe it would be good to formally define this concept that is central to the paper.

L.96 Is Thomson et al., 2011 the correct reference? The sentence was about great tits, the paper is about trouts.

L.105 Clarify what integration looks like in term of G properties.

L.145-147 "We also predict that G will be dominated by a single major axis of genetic variation in multivariate trait space, but are more circumspect about how that might look." is a vague and problematic prediction; in fact I think it is the weakest point of the paper. What does define a single major axis? Would you consider this prediction correct if you had one axis explaining 51% of variation and another one 49%? What about one axis at 49% and others around 10%? Is the absolute proportion captured by the first axis what matters, or is it the difference between the proportion captured by different axes? What values should the absolute or relative proportions captured by the first axis take for your prediction to be confirmed? What is the null distribution of joint-proportions captured by each axis and do empirical estimates deviate from that null? Without a clearer expectation I find the prediction not very useful and it is difficult to interpret the eigen decomposition of G (Figure 4). If a clear prediction cannot be formulated I think the author should refrain from making a prediction at all and instead say they will describe the pattern of eigen components.

L.335 Similarly: "This structure is suggestive of a single major axis…" Maybe, but without a definition of what defines a "single major axis", this conclusion is void.

L.514 "Our results suggest that correlational selection in the past has likely shaped the multivariate stress response". I think this conclusion is incorrect and unrelated to the results. The G matrix is not only shaped by selection, but also by mutational input (in your case it is not difficult to imagine that mutations have pleiotropic effects on behaviour and cortisol levels), migration (and drift). In itself, an instantaneous picture of G says little about processes that have shaped it in the past.

---

## [Author Response]

Reviewer #1 (Recommendations for the authors):Overall, I thought this paper was fantastic. My public review reflects my enthusiasm for the work. Most of my points below amount to writing suggestions.My only substantive comment is that I think it would be helpful to show a table of all the eigenvectors and eigenvalues, not just the figure showing the first eigenvector loadings. This could be put into the supplement with the other tables.

We are very grateful to Reviewer 1 for their positive comments, and we respond point-by-point below. As requested, we have included a supplemental table of the eigen vectors and values (along with a supplemental figure of the proportion of variance explained by each vector, with 95% confidence intervals from a bootstrapping procedure).

70-71: Need some more citations here: Maybe Cheverud 1982 (which is cited later), Roff and Fairbairn 2012

Thank you for the suggestions – now inserted.

143: This is just a style preference, but I always prefer past tense when it comes to the word "predict" in the introduction. By the time the paper is being written, the prediction is in the past. There are some places in the results like this where it is written in present tense but seems like it should be in past tense.

Agreed, and we have made several edits towards the end of the introduction accordingly.

146-147: This sentence is perhaps too informal.

Now changed to “We also predicted that both behavioural and endocrine components of the stress response would load on the major axis of genetic variation in multivariate trait space, g_max_.”

162: Multivariate doesn't need to be in parentheses.

Parentheses removed.

314-327: The organization of this paragraph could be improved a bit. It was tough to follow the results and I had to read it over again to get it.

We have made edits to this paragraph, and hope that its readability has improved.

Figure 3: Would these make more sense with cortisol on the X? I know it's not a regression, but people tend to think of the path hormone -> behavior.

We have updated Figure 3 to put cortisol on the x axis (and to include a new panel for the genetic relationship between cortisol and freezings, in response to suggestions from other reviewers).

479: I think it might be useful to have one more logical step linking the results to potential past correlational selection. Specifically, some explanation about how these traits might interact to influence fitness would help. It's not entirely clear whether the authors are envisioning correlational selection on a suite of behaviors or correlational selection on hormone levels plus one or more of the behaviors. I'm obviously sympathetic to the idea that correlational selection influences hormone-mediated suites, but I do worry that others might see discussing of it here as too speculative (see below). The authors don't necessarily need to talk about it to explain the significance of their results.

We have inserted some text about the assumptions here as regards selection favouring combinations of trait values that yield higher fitness, and that our interpretation is necessarily speculative.

507: I may be misunderstanding this, but wouldn't the reactive-proactive model be consistent with a single axis as well? Reactive would be on one end of the spectrum and proactive would be on the other. This goes back to lines 150-159. The way it is described sounds like a single axis. Is the difference just that the two ends of the continuum are more extreme in the reactive-proactive model?

We think this was just a slight misunderstanding of our intended meaning (suggesting that we were not clear enough in our original version), however this section has now been rewritten in the light of new results including the number of freezings (as advised by reviewer 2).

514: Following on the comment above, I like the idea of correlational selection explaining the effect, but it might be too much of a speculation to lead off the last paragraph of the paper with it. In other words, I think the results are consistent with it, but don't quite suggest it.

We agree and have removed this (as well as caveated other sections that reference correlational selection) – please see response to reviewer 3 where we address this in more detail.

Reviewer #3 (Recommendations for the authors):Title: I am not sure what "evolutionary genetic integration" means. "(Quantitative) Genetic integration" sure, but what does the "evolutionary" add? All biological states are the product of evolution, so in a sense "evolutionary" could apply to all biological objects but be quite pointless. The study describes the current state of G in the population but does not describe an evolutionary process (past or on-going).

“Evolutionary genetic” is often used in our field in the context of polygenic traits studied within an explicitly evolutionary framework; however, we agree for a general audience this is not particularly meaningful. We have therefore removed ‘evolutionary’ from the title (now ‘Genetic integration of the behavioural and endocrine components of the stress response’).

L.69-73 I agree that past correlational selection would have tended to shape the G matrix along a major axis, but it is difficult to predict a priori what a G matrix should look like if you do not know the variance-covariance of mutational (and migration) input was or what were the exact properties of correlational selection. So "should" we really expect genetic and phenotypic integration? The expectation is not clear to me. Maybe make clearer what the expectation assumes.

We have modified the text here to be more circumspect about this point, and highlighted relevant assumptions about mutational input:

“Evolutionary theory predicts that correlational selection will shape the structure of multivariate quantitative genetic variance (as represented by the genetic covariance matrix G; Cheverud 1982; Blows 2007; Roff and Fairbairn 2012), contingent on the distribution of new pleiotropic mutations that generate multivariate genetic variance as well as the selection that depletes it (Blows and Walsh 2009; Walsh and Blows 2009). In general, correlational selection should (under certain assumptions) have direct and strong effects on genetic covariances (Lande 1980; Jones, Arnold and Bürger 2003; Arnold *et al.,* 2008). In the context of the stress response we should therefore expect genetic – as well as phenotypic – integration of behavioural and physiological traits (McGlothlin and Ketterson 2008; Ketterson, Atwell and McGlothlin 2009; Cox, McGlothlin and Bonier 2016).”

L.72 First occurrence of the word "integration" in the main text. Maybe it would be good to formally define this concept that is central to the paper.

We have added the following sentence here:

“By genetic integration, we mean genetic correlation structure among suites of traits. Genetic integration of the stress response components is hypothesised to underpin adaptive variation in multivariate phenotypes in many fields of evolutionary biology (e.g., life history (Stearns 1989; Roff 1992), behavioural ecology (Sih and Bell 2008)), but has not been tested for explicitly using quantitative genetic approaches.”

L.96 Is Thomson et al., 2011 the correct reference? The sentence was about great tits, the paper is about trouts.

Thank you for catching this – the sentence was also meant to refer to trout (as per the citation). Amended to reflect this.

L.105 Clarify what integration looks like in term of G properties.

We have inserted the following sentence here:

“Integration in G is manifest as genetic correlations between trait pairs, and also at the level of the whole matrix by an overdispersion of eigenvalues—indicating that there are fewer effective dimensions of genetic variation than there are traits measured (Blows 2007).”

L.145-147 "We also predict that G will be dominated by a single major axis of genetic variation in multivariate trait space, but are more circumspect about how that might look." is a vague and problematic prediction; in fact I think it is the weakest point of the paper. What does define a single major axis? Would you consider this prediction correct if you had one axis explaining 51% of variation and another one 49%? What about one axis at 49% and others around 10%? Is the absolute proportion captured by the first axis what matters, or is it the difference between the proportion captured by different axes? What values should the absolute or relative proportions captured by the first axis take for your prediction to be confirmed? What is the null distribution of joint-proportions captured by each axis and do empirical estimates deviate from that null? Without a clearer expectation I find the prediction not very useful and it is difficult to interpret the eigen decomposition of G (Figure 4). If a clear prediction cannot be formulated I think the author should refrain from making a prediction at all and instead say they will describe the pattern of eigen components.

We accept this as a valid criticism. To our knowledge there are no formal definitions of what comprises a ‘single’ major axis and so we are reluctant to introduce one de novo (especially in a *post hoc* context). We have therefore amended the statement to emphasis that our goal is to describe the patterns. Nevertheless, we do think that it is appropriate to provide some predictions—even if qualitative—to help a reader unfamiliar with eigen decomposition of covariance matrices to understand the connection to the biology we describe. Rather than predicting a ‘single’ major axis, we now explicitly predict that the major axis of genetic variation (ie, g_max_) will load on both behavioural and endocrine components (“We also predicted that both behavioural and endocrine components of the stress response would load on the major axis of genetic variation in multivariate trait space, g_max_.”). We return later to the theme of just how dominant this axis is, using the confidence intervals around the eigen values in our post hoc interpretation of results.

L.335 Similarly: "This structure is suggestive of a single major axis…" Maybe, but without a definition of what defines a "single major axis", this conclusion is void.

Agreed – we have removed ‘single’ but the point (at least, as we meant it) still stands that the major axis of genetic variation (i.e., PC1 or g_max_) describes multivariate genetic variation in integrated stress response because both behavioural and endocrine components load on it.

L.514 "Our results suggest that correlational selection in the past has likely shaped the multivariate stress response". I think this conclusion is incorrect and unrelated to the results. The G matrix is not only shaped by selection, but also by mutational input (in your case it is not difficult to imagine that mutations have pleiotropic effects on behaviour and cortisol levels), migration (and drift). In itself, an instantaneous picture of G says little about processes that have shaped it in the past.

We agree that the first part of this sentence was overstating the conclusions that can be drawn from our results, and have removed the clause about correlational selection here. We also edited the earlier line on this topic to be explicit that our results are consistent with but not proof of the idea that correlational selection in the past has led to coevolution of these stress components.